# Going Beyond Heuristics by Imposing Policy Improvement as a Constraint

**Chi-Chang Lee**[1*]**, Zhang-Wei Hong**[2*] **, Pulkit Agrawal**[2]
Improbable AI Lab
Massachusetts Institute of Technology

## Abstract

In many reinforcement learning (RL) applications, incorporating heuristic rewards alongside the task reward is crucial for achieving desirable performance. Heuristics encode prior human knowledge about how a task should be done, providing valuable hints for RL algorithms. However, such hints may not be optimal, limiting the performance of learned policies. The currently established way of using heuristics is to modify the heuristic reward in a manner that ensures that the optimal policy learned with it remains the same as the optimal policy for the task reward (i.e., optimal policy invariance). However, these methods often fail in practical scenarios with limited training data. We found that while optimal policy invariance ensures convergence to the best policy based on task rewards, it doesn't guarantee better performance than policies trained with biased heuristics under a finite data regime, which is impractical. In this paper, we introduce a new principle tailored for finite data settings. Instead of enforcing optimal policy invariance, we train a policy that combines task and heuristic rewards and ensures it outperforms the heuristic-trained policy. As such, we prevent policies from merely exploiting heuristic rewards without improving the task reward. Our experiments on robotic locomotion, helicopter control, and manipulation tasks demonstrate that our method consistently outperforms the heuristic policy, regardless of the heuristic rewards' quality. Code is available at https://github.com/Improbable-AI/hepo.

## 1 Introduction

Reinforcement learning (RL) [1] is a powerful framework for learning policies that can surpass human performance in complex tasks. However, training RL policies with sparse or delayed rewards is often ineffective. Instead of relying solely on sparse task rewards that indicate an agent's success or failure, it is common to augment the sparse reward with *heuristic* reward terms that provide denser reward supervision to speed up and improve the performance of RL policies [2, 3]. Shining examples of the success and necessity of heuristic reward terms are complex robotic object manipulation [4] and locomotion [5–8] tasks. However, heuristics impose human assumptions that may limit the RL algorithm. For example, a heuristic reward function may encourage a robot to walk like a human, yet there could be faster walking policies that don't resemble human gait.

The key question is how to learn a policy $\pi$ that outperforms one trained solely on heuristics (i.e., heuristic policy $\pi_H$). Practitioners tackle this problem by tuning the balance between the task objective $J(\pi)$ and the heuristic objective $H(\pi)$ in the augmented training objective $J(\pi) + \lambda H(\pi)$, where $\lambda$ controls the balance between the two objectives for a policy $\pi$. However, it requires careful tuning of $\lambda$ to make the policy $\pi$ outperform the heuristic policy; otherwise, the algorithm might prioritize heuristic rewards while neglecting the task objective.

---

[*]indicates equal contribution. [1] National Taiwan University, Taiwan. [2] Improbable AI Lab, MIT, Cambridge, USA.

38th Conference on Neural Information Processing Systems (NeurIPS 2024).

Tuning $\lambda$ to balance both objectives is time-consuming. We desire an algorithm that finds a policy that outperforms the heuristic policy without requiring such tuning for any given heuristic reward function. Classic methods [2, 9–14] modify heuristic rewards to align the augmented objective's optimal policy with the one for the task objective (i.e., optimal policy invariance), theoretically ensuring that with infinite data the policy outperforms the heuristic. However, in practice, these modified heuristics often fall short on complex robotic tasks compared to policies trained solely on heuristic objectives, as demonstrated in our study (Section 4.1) and prior work [10].

In this paper, we challenge the prevailing paradigm by questioning whether optimal policy invariance is the appropriate objective to prevent heuristics from limiting RL agent's performance in the finite data regime. As optimal policy invariance ensures convergence to the optimal policy with infinite data it may not be practical in many real-world settings. We propose an alternative paradigm that, in every step of training, imposes the constraint of improving task performance beyond a policy trained solely on heuristic rewards (i.e., $J(\pi) \geq J(\pi_H)$). This condition $J(\pi) \geq J(\pi_H)$ guarantees that the learned policy $\pi$ outperforms the heuristic policy $\pi_H$, effectively surpassing human-designed heuristics. Additionally, such policy improvements can be verified and achieved using many existing deep RL algorithms [15–18] in finite data settings.

Therefore, we enforce the policy improvement condition $J(\pi) \geq J(\pi_H)$ as a constraint, preventing the policy from exploiting heuristic rewards during training. We propose the following constrained optimization objective:

$$\max_{\pi} J(\pi) + H(\pi) \quad \text{subject to} \quad J(\pi) \geq J(\pi_H)$$

Optimizing this objective at each iteration allows learning a policy performing better than or equal to policies trained only on heuristic rewards. It prevents capitalizing on heuristic rewards at the expense of task rewards. Moreover, it enables adaptively balancing both rewards over time instead of using a fixed coefficient $\lambda$. Our contribution is an add-on to existing deep RL algorithms to improve RL algorithms trained with heuristic rewards. We evaluated our method on robotic locomotion, helicopter, and manipulation tasks using the IsaacGym simulator [19]. The results show that our method led to superior task rewards and higher task-completion success rates compared to the policies solely trained with heuristic rewards, even when heuristic rewards are ill-designed.

## 2  Preliminaries: Reinforcement Learning with Heuristic

**Reinforcement Learning (RL):**   RL is a popular paradigm for solving sequential decision-making problems [1] where the problems are modeled as an interaction between an agent and an unknown environment [1]. The agent aims to improve its performance through repeated interactions with the environment. At each round of interaction, the agent starts from the environment's initial state $s_0$ and samples the corresponding trajectory. At each timestep $t$ within that trajectory, the agent perceives the state $s_t$, takes an action $a_t \sim \pi(.|s_t)$ according to its policy $\pi$, receives a *task* reward $r_t = r(s_t, a_t)$, and transitions to a next state $s_{t+1}$ until reaching terminal states, after which a new trajectory is initialized from $s_0$, and the cycle repeats. The agent's goal is to learn a policy $\pi$ that maximizes the expected return $J(\pi)$ in a trajectory as below:

$$J(\pi) = \mathbb{E}_{\pi}[\sum_{t=0}^{\infty} \gamma^t r(s_t, a_t)], \tag{1}$$

where $\gamma$ denotes a discount factor [1] and $\mathbb{E}_{\pi}[.]$ denotes taking expectation over the trajectories sampled by $\pi$. In the following, we term $J$ as the true *task* objective, as it indicates the performance of a policy on the task.

**RL with Heuristic:**   In many tasks, learning a policy to maximize the true objective $J$ is challenging because rewards may be sparse or delayed. This lack of feedback makes policy optimization difficult for RL algorithms. To address this, practitioners often use a heuristic reward function $h$ with denser reward signals to facilitate optimization, aiming to learn a policy that performs better in $J$. The policy trained to maximize the expected return of heuristic rewards is called the *heuristic* policy $\pi_H$. The expected return of heuristic rewards, termed the *heuristic* objective $H$, is defined as:

$$H(\pi_H) = \mathbb{E}_{\pi_H} \left[ \sum_{t=0}^{\infty} \gamma^t h(s_t, a_t) \right], \tag{2}$$

where $h(s_t, a_t)$ is the heuristic reward at timestep $t$ for state $s_t$ and action $a_t$.

# 3 Method: Improving Heuristic Policy via Constrained Optimization

**Problem statement:** Optimizing both task $J$ and heuristic $H$ objectives jointly could lead to better task performance than training solely with $J$ or $H$, but needs careful tuning on the weight coefficient $\lambda$ among both objectives in $\max_\pi J(\pi) + \lambda H(\pi)$. Without careful tuning, the policy $\pi$ may learn to exploit heuristic rewards $H$ and compromise performance of $J$. The goal of this paper is to mitigate the requirement of tuning this coefficient to balance them in order to improve task performance.

**Key insight - Leveraging Heuristic with Constraint:** We aim to use the heuristic objective $H$ for training only when it improves task performance $J$ and ignore it otherwise. Rather than manually tuning the weight coefficient $\lambda$ to balance both rewards, we introduce a key insight: impose a *policy improvement* constraint (i.e., $J(\pi) \geq J(\pi_H)$) during training. This prevents RL algorithms from exploiting heuristic rewards $H$ at the expense of task rewards $J$. To achieve this goal, we introduce the following constrained optimization objective:

$$\max_\pi J(\pi) + H(\pi) \text{ subject to } J(\pi) \geq J(\pi_H). \tag{3}$$

This constrained objective (Equation 3) results in an improved policy $\pi$ over the heuristic policy $\pi_H$, leading us to call this framework *Heuristic-Enhanced Policy Optimization (HEPO)*. A practical algorithm to optimize this objective is presented in Section 3.1, and its implementation on a widely-used RL algorithm [15] in robotics is detailed in Section 3.2.

## 3.1 Algorithm: Heuristic-Enhanced Policy Optimization (HEPO)

Finding feasible solutions for the constrained optimization problem in Equation 3 is challenging due to the nonlinearity of the objective function $J$ with respect to $\pi$. One practical approach is to convert it into the following unconstrained min-max optimization problem using Lagrangian duality:

$$\min_{\alpha \geq 0} \max_\pi \mathcal{L}(\pi, \alpha), \text{ where } \mathcal{L}(\pi, \alpha) := J(\pi) + H(\pi) + \alpha \left( J(\pi) - J(\pi_H) \right), \tag{4}$$

where the Lagrangian multiplier is $\alpha \in \mathbb{R}^+$. We can optimize the policy $\pi$ and the multiplier $\alpha$ for this min-max problem by a gradient descent-ascent strategy, alternating between optimizing $\pi$ and $\alpha$.

**Enhanced policy $\pi$:** The optimization objective for the policy $\pi$ can be obtained by rearranging Equation 4 as follows:

$$\max_\pi \ (1 + \alpha)J(\pi) + H(\pi),$$

$$\text{where } (1 + \alpha)J(\pi) + H(\pi) = \mathbb{E}_\pi \left[ \sum_{t=0}^\infty \gamma^t \big( (1 + \alpha)r(s_t, a_t)) + h(s_t, a_t) \big) \right]. \tag{5}$$

This represents an unconstrained regular RL objective with the modified reward at each step as $(1 + \alpha)r(s_t, a_t) + h(s_t, a_t)$, which can be optimized using any off-the-shelf deep RL algorithm. In this modified reward, the task reward $r(s_t, a_t)$ is weighted by the Lagrangian multiplier $\alpha$, reflecting the potential variation in the task reward's importance during training as $\alpha$ evolves. The interaction between the update of the Lagrangian multiplier and the policy will be elaborated upon next.

**Lagrangian Multiplier $\alpha$:** The Lagrangian multiplier $\alpha$ is optimized for Equation 4 by stochastic gradient descent, with the gradient defined as:

$$\nabla_\alpha \mathcal{L}(\pi, \alpha) = J(\pi) - J(\pi_H). \tag{6}$$

Notably, $\nabla_\alpha \mathcal{L}(\pi, \alpha)$ is exactly the performance gain of the policy $\pi$ over the heuristic policy $\pi_H$ on the task objective $J$. By applying gradient descent with $\nabla_\alpha \mathcal{L}(\pi, \alpha)$, when $J(\pi) > J(\pi_H)$ and thus $\nabla_\alpha \mathcal{L}(\pi, \alpha) > 0$, the Lagrangian multiplier $\alpha$ decreases. As $\alpha$ represents the weight of the task reward in Equation 5, it indicates that when $\pi$ outperforms $\pi_H$, the importance of the task reward diminishes because $\pi$ already achieves superior performance compared to the heuristic policy $\pi_H$ regarding the task objective $J$. Conversely, when $J(\pi) < J(\pi_H)$, $\alpha$ increases, thereby emphasizing the importance of task rewards in optimization. The update procedure for the Lagrangian multiplier $\alpha$ offers an adaptive reconciliation between the heuristic reward $h$ and the task reward $r$.

## 3.2 Implementation

We present a practical approach to optimize the min-max problem in Equation 4 using Proximal Policy Optimization (PPO) [15]. We selected PPO because it is widely used in robotic applications involving heuristic rewards, although our HEPO framework is not restricted to PPO. The standard PPO implementation involves iterative stochastic gradient descent updates over numerous iterations, alternating between collecting trajectories with policies and updating those policies. We outline the optimization process for each iteration and provide a summary of our implementation in Algorithm 1.

**Training policies $\pi$ and $\pi_H$:** Instead of pre-training the heuristic policy $\pi_H$, which requires additional data and reduces data efficiency, we concurrently train both the enhanced policy $\pi$ and the heuristic policy $\pi_H$, allowing them to share data. For each iteration $i$, we gather trajectories $\tau$ and $\tau_H$ using the enhanced policy $\pi^i$ and the heuristic policy $\pi_H^i$, respectively. Following PPO's implementation, we compute the advantages $A_r^{\pi^i}(s_t, a_t)$, $A_r^{\pi_H^i}(s_t, a_t)$, $A_h^{\pi^i}(s_t, a_t)$, and $A_h^{\pi_H^i}(s_t, a_t)$ for the task reward $r$ and heuristic reward $h$ with respect to $\pi^i$ and $\pi_H^i$. We then weight the advantage with the action probability ratio between the new policies being optimized (i.e., $\pi^{i+1}$ and $\pi_H^{i+1}$) and the policies collecting the trajectories (i.e., $\pi^i$ or $\pi_H^i$). Finally, we optimize the policies at the next iteration $i+1$ for the objectives in Equations 7 and 8:

$$\pi^{i+1} \leftarrow \arg\max_{\pi} \mathbb{E}_{\tau \sim \pi^i} \left[ \frac{\pi(a_t|s_t)}{\pi^i(a_t|s_t)} \left( (1+\alpha) A_r^{\pi^i}(s_t, a_t) + A_h^{\pi^i}(s_t, a_t) \right) \right] + \tag{7}$$

$$\mathbb{E}_{\tau_H \sim \pi_H^i} \left[ \frac{\pi(a_t|s_t)}{\pi_H^i(a_t|s_t)} \left( (1+\alpha) A_r^{\pi_H^i}(s_t, a_t) + A_h^{\pi_H^i}(s_t, a_t)+ \right) \right] \quad \text{(Enhanced policy)}$$

$$\pi_H^{i+1} \leftarrow \arg\max_{\pi} \mathbb{E}_{\tau_H \sim \pi_H^i} \left[ \frac{\pi(a_t|s_t)}{\pi_H^i(a_t|s_t)} A_h^{\pi^i}(s_t, a_t) \right] + \tag{8}$$

$$\mathbb{E}_{\tau \sim \pi^i} \left[ \frac{\pi(a_t|s_t)}{\pi^i(a_t|s_t)} A_h^{\pi_H^i}(s_t, a_t) \right] \quad \text{(Heuristic policy)}.$$

Maximizing the advantages will result in a policy that maximizes the expected return for a chosen reward function, as demonstrated in PPO [15]. This enables us to maximize the objective $J$ and $H$. We estimate the advantages $A_r^{\pi^i}(s_t, a_t)$ and $A_h^{\pi^i}(s_t, a_t)$ (or $A_r^{\pi_H^i}(s_t, a_t)$ and $A_h^{\pi_H^i}(s_t, a_t)$) using the standard PPO implementation with different reward functions. Therefore, we omit the details of the advantage's clipped surrogate objective in PPO, and leave them in Appendix A.1.

Although PPO is an on-policy algorithm, the use of off-policy importance ratio correction (i.e., the action probability ratios between two policies) allows us to use states and actions generated by another policy. This enables us to train $\pi$ using data from $\pi_H$ and vice versa. Both policies $\pi$ and $\pi_H$ are trained using the same data but with different reward functions. Note that collecting trajectories from both policies does not require more data than the standard PPO implementation. We collect half the trajectories with each policy, $\pi$ and $\pi_H$, for a total of $B$ trajectories (see Algorithm 1). Then, we update both $\pi$ and $\pi_H$ using all $B$ trajectories.

**Optimizing the Lagrangian multiplier $\alpha$:** To update the Lagrangian multiplier $\alpha$, we need to compute the gradient in Equation 6, which corresponds to the performance gain of the enhanced policy $\pi$ over the heuristic policy $\pi_H$ on the task objective $J$. Utilizing the performance difference lemma [20, 16], we relate this improvement to the expected advantages over trajectories sampled by the enhanced policy $\pi$ as $J(\pi) - J(\pi_H) = \mathbb{E}_\pi [A^{\pi_H} r(s_t, a_t)]$. However, this approach only utilizes half of the trajectories at each iteration since it exclusively relies on trajectories from the enhanced policy $\pi$. To leverage trajectories from both policies, we also consider the performance gain in the reverse direction as $-(J(\pi_H) - J(\pi)) = -\mathbb{E}_{\pi_H} [A^\pi r(s_t, a_t)]$. Consequently, we can estimate the gradient of $\alpha$ using trajectories from both policies, as illustrated below:

$$\nabla_\alpha \mathcal{L}(\pi, \alpha) = J(\pi) - J(\pi_H) = \mathbb{E}_\pi [A_r^{\pi_H}(s_t, a_t)] \tag{9}$$

$$= -(J(\pi_H) - J(\pi)) = -\mathbb{E}_{\pi_H} [A_r^\pi(s_t, a_t)]. \tag{10}$$

At each iteration $i$, we estimate the gradient of $\alpha$ using the advantage $A_r^{\pi_H}(s_t, a_t)$ and $A_r^\pi(s_t, a_t)$ on the trajectories sampled from both $\pi^i$ and $\pi_H^i$, and update $\alpha$ with stochastic gradient descent as follows:

$$\alpha \leftarrow \alpha - \frac{\eta}{2} \left( \mathbb{E}_{\tau \sim \pi^i} \left[ A_r^{\pi_H^i}(s_t, a_t) \right] - \mathbb{E}_{\tau \sim \pi_H^i} \left[ A_r^{\pi^i}(s_t, a_t) \right] \right), \tag{11}$$

where $\eta \in \mathbb{R}^+$ is the step size. The expected advantage in Equation 11 are estimated using the generalized advantage estimator (GAE) [21].

---

**Algorithm 1** Heuristic-Enhanced Policy Optimization (HEPO)

---

1: **Input:** Number of trajectories per iteration $B$
2: Initialize the enhanced policy $\pi^0$, the heuristic policy $\pi_H^0$, and the Lagrangian multiplier $\alpha$
3: **for** $i = 0 \cdots$ **do**                                             $\triangleright$ $i$ denotes iteration index
4:     Rollout $B/2$ trajectories $\tau$ by $\pi^i$
5:     Rollout $B/2$ trajectories $\tau_H$ by $\pi_H^i$
6:     $\pi^{i+1} \longleftarrow$ Train the policy $\pi^i$ for optimizing Equation 7 using both $\tau$ and $\tau_H$
7:     $\pi_H^{i+1} \longleftarrow$ Train the policy $\pi_H^i$ for optimizing Equation 8 using both $\tau$ and $\tau_H$
8:     $\alpha \longleftarrow$ Update the Lagrangian multiplier $\alpha$ by gradient descent (Equation 9) using $\tau$ and $\tau_H$
9: **end for**

---

### 3.3 Connection to Extrinsic-Intrinsic Policy Optimization [22]

Closely related to our HEPO framework, Chen et al. [22] proposes Extrinsic-Intrinsic Policy Optimization (EIPO), which trains a policy to maximize both task rewards and exploration bonuses [23] subject to the constraint that the learned policy $\pi$ must outperform the *task* policy $\pi_J$ trained solely on task rewards. HEPO and EIPO differ in their objective functions and implementation of the constrained optimization problem. Additional information can be found in the Appendix, covering the objective formulation (Appendix A.1), implementation tricks (Appendix A.2), and detailed pseudocode (Appendix A.3).

Exploration bonuses [23] can be viewed as heuristic rewards. The main difference between HEPO and EIPO's optimization objectives lies in constraint design. Both frameworks require the learned policy $\pi$ to outperform a reference policy $\pi_{\text{ref}}$ (i.e., $J(\pi) \geq J(\pi_{\text{ref}})$) but use a different reference policy. EIPO uses the task policy $\pi_J$ as the reference policy $\pi_{\text{ref}}$ because they aim for asymptotic optimality in task rewards. If the constraint is satisfied with $\pi_J$ being the optimal policy for task rewards, the learned policy $\pi$ will also be optimal for task rewards. In contrast, HEPO uses the heuristic policy $\pi_H$ trained solely on heuristic rewards since HEPO aims to improve upon it.

HEPO simplifies the implementation. Both HEPO and EIPO train two policies with shared data, but EIPO alternates the policy used for trajectory collection each iteration and has a complex switching rule, which introduces more hyperparameters. HEPO collects trajectories using both policies together at each iteration, simplifying implementation and avoiding extra hyperparameters.

## 4 Experiments

We evaluate whether HEPO enhances the performance of RL algorithms in maximizing task rewards while training with heuristic rewards. We conduct experiments on 9 tasks from IsaacGym (ISAAC) [19] and 20 tasks from the Bidexterous Manipulation (BI-DEX) benchmark [24]. These tasks rely on heavily engineered reward functions for training RL algorithms. Each task has a task reward function $r$ that defines the task objective $J$ to be maximized, and a heuristic reward function $h$ that defines the heuristic objective $H$, provided in the benchmarks to facilitate the optimization of task objectives $J$. We implement HEPO based on PPO [15] and compare it with the following baselines:

- **H-only (heuristic only)**: This is the standard PPO baseline provided in ISAAC. The policy is trained solely using the heuristic reward: $\max_\pi H(\pi)$. The heuristic reward functions in ISAAC and BI-DEX are designed to help RL algorithms maximize the task objective $J$. This baseline is crucial to determine if an algorithm can surpass a policy trained with highly engineered heuristic rewards.

- **J-only (task only)**: The policy is trained using only the task reward: $\max_\pi J(\pi)$. This baseline demonstrates the performance achievable without heuristics. Ideally, algorithms that incorporate heuristics should outperform this baseline.

- **J+H (mixture of task and heuristic)**: The policy is trained using a mixture of task and heuristic rewards: $\max_\pi J(\pi) + \lambda H(\pi)$, with $\lambda$ balancing the two rewards. As [22] shows, proper tuning of $\lambda$ can enhance task performance by balancing both training objectives.

- **Potential-based Reward Shaping (PBRS) [2]**: The policy is trained to maximize $\mathbb{E}_\pi[\sum_{t=0}^{\infty} \gamma^t r_t + \gamma h_{t+1} - h_t]$, where $r_t$ and $h_t$ are the task and heuristic rewards at timestep $t$. PBRS guarantees that the optimal policy is invariant to the task reward function. We include it as a baseline to examine if these theoretical guarantees hold in practice.

- **HuRL [10]**: The policy is trained to maximize $\mathbb{E}_\pi[\sum_{t=0}^{\infty} \gamma^t r_t + (1 - \beta_i)\gamma h_{t+1}]$, where $\beta_i$ is a coefficient updated at each iteration to balance heuristic rewards during different training stages. The scheduling mechanism is detailed in [10] and our source code provided in the Supplementary Material.

- **EIPO [22]**: The policy is trained using the constrained objective: $\max_\pi J(\pi) + H(\pi)$ s.t. $J(\pi) \geq J(\pi_J)$, where $\pi_J$ is the policy trained with task rewards only. EIPO is similar to HEPO but differs in formulation and implementation, as detailed in Section 4.3.

Each method is trained for 5 random seeds and implemented based on the open-sourced implementation [25], where the detailed training hyperparameters can be found in Appendix A.4.

**Metrics:** Based on the task success criteria in ISAAC and BI-DEX, we consider two types of task reward functions $r$: (i) Progressing (for locomotion or helicopter robots) and (ii) Goal-reaching (for manipulation). In progressing tasks, robots aim to maximize their traveling distance or velocity from an initial point to a destination. Thus, movement progress is defined as the task reward. In goal-reaching tasks, robots aim to complete assigned goals by reaching specific goal states. Here, task rewards are binary, with a value of $1$ indicating successful attainment of the goal and $0$ otherwise. Detailed descriptions of our task objectives and total reward definitions are provided in Appendix C.

## 4.1 Benchmark results

**Setup:** We aim to determine if HEPO achieves higher task returns and improves upon the policy trained with only heuristic rewards (H-only) in the majority of tasks. In this experiment, we use the heuristic reward functions from the ISAAC and BI-DEX benchmarks. To measure performance improvement over the heuristic policy, we normalize the return of each algorithm $X$ using the formula $(J_X - J_{\text{random}})/(J_{\text{H-only}} - J_{\text{random}})$, where $J_X$, $J_{\text{H-only}}$, and $J_{\text{random}}$ denote the task returns of algorithm $X$, the heuristic policy, and the random policy, respectively. In Figure 1, we present the interquartile mean (IQM) of the normalized return and the probability of improvement for each method across 29 tasks, following [26]. IQM, also known as the 25% trimmed mean, is a robust estimate against outliers. It discards the bottom and top 25% of runs and calculates the mean score of the remaining 50%. The probability of improvement measures whether an algorithm performs better than another, regardless of the margin of improvement. Both approaches prevent outliers from dominating the performance estimate.

**Results:** The results in Figure 1 indicate that policies trained with task rewards only (J-only) generally perform worse than those trained with heuristics, both in terms of IQM of normalized return and probability of improvement. PBRS does not improve upon J-only, demonstrating that the optimal policy invariance guarantee rarely holds in practice. Both EIPO and HuRL outperform J-only but do not surpass H-only, demonstrating that neither approach can improve upon the heuristic policy. Policies trained with both task and heuristic rewards (J+H) perform slightly worse than those trained with heuristics only (H-only), possibly because the weight coefficient balancing both rewards is too task-sensitive to work across all tasks. HEPO, however, outperforms all other methods in both IQM of normalized returns and shows a probability of improvement over the heuristic policy greater than 50%, indicating statistically significant improvements as suggested by Agarwal et al. [26]. Complete learning curves are presented in the Appendix B.1. Additional results on various benchmarks and RL algorithms are provided in Appendix B.3, demonstrating that HEPO is effective in hard-exploration tasks using exploration bonuses [23] and with RL algorithms beyond PPO.

## 4.2 Can HEPO be robust to reward functions designed in the wild?

**Setup:** We envision to develop an RL algorithm that can effectively utilize heuristic reward functions, thereby reducing the time costs associated with reward design. We simulate reward design scenarios

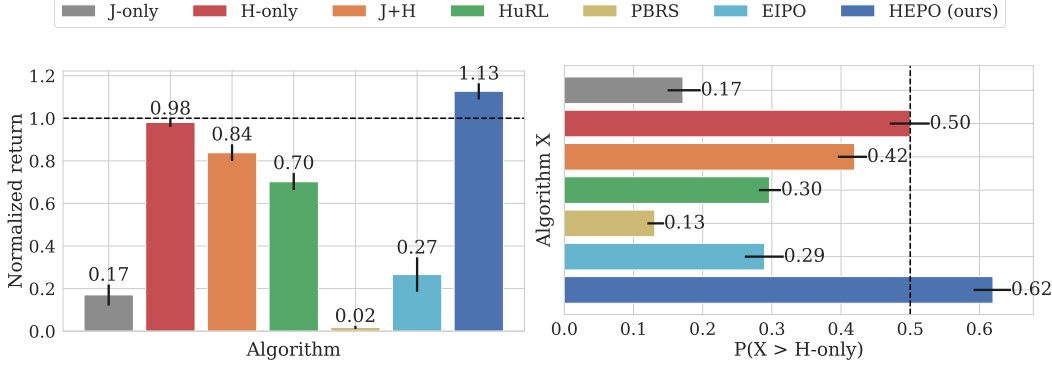

(a) IQM over ISAAC + BI-DEX  (b) Probability of Improvement over ISAAC + BI-DEX

Figure 1: **(a)** The vertical axis represents the interquartile mean (IQM) [26] of normalized task return across 29 tasks. HEPO outperforms the policy trained solely with a heavily engineered heuristic reward function (H-only) and other methods, demonstrating that HEPO makes better use of heuristic rewards for learning. **(b)** The horizontal axis shows the probability that algorithm X outperforms the policy trained solely with heuristic rewards (H-only). HEPO achieves a 62% probability of improvement over the heuristic policy on average, with the lower bound of the confidence interval above 50%, indicating statistically significant improvements over the heuristic policy.

and evaluate the algorithm's performance when trained with heuristic reward functions created under real-world conditions. Unlike the highly engineered reward functions in ISAAC, participants were asked to design their heuristic reward functions within a short time frame. This approach assesses the algorithm's effectiveness when trained with less refined heuristic reward functions. Participants were asked to iterate on their reward design by writing the reward function, training a policy with a given RL algorithm, reviewing the videos and learning curves, refining the reward, and repeating the process. We selected the *FrankaCabinet* task for this study because its original heuristic reward function is heavily engineered and consisting of many terms. The task of *FrankaCabinet* is training a robot arm to open a cabinet. We recruited twelve graduate students with varying levels of proficiency in machine learning and robotics and divided them into two groups. One group used HEPO to iterate on heuristic rewards, while the other group used PPO. This approach ensures that the designed heuristic reward functions are not specialized for one algorithm and ineffective for another. Each participant was instructed to edit the heuristic reward function to help RL algorithms maximize the task return. We used the same task reward metric as in Section 4.1. We then used the final versions of their heuristic reward functions to train HEPO and PPO, and reported the normalized return of the learned policies. Note that we adhered to the normalization scheme outlined in Section 4.1, based on the performance of the policy trained with the original heuristic reward function. This approach allows us to observe any performance drop when training policies with less engineered heuristic reward functions.

**Quantitative results:** Figure 2 shows that HEPO achieves significantly higher (lower confidence bound above the baselines' upper confidence bound) average normalized returns than PPO trained only on heuristic rewards (PPO (H-only)) in 9 out of 12 heuristic reward functions. Additionally, Table 1 indicates that across all heuristic functions, HEPO achieves a higher interquartile mean (IQM) of normalized returns and has a statistically significant probability of outperforming PPO (H-only) with lower confidence bound greater than 0.5. This suggests that even when trained with poorly designed heuristic reward functions, HEPO performs better than PPO (H-only). Notably, PPO (H-only) that is trained with $H2$, $H5$, and $H6$ achieves normal-

| Algo. X | IQM | P(X > H-only) |
|---------|-----|---------------|
| H-only | 0.44 (0.37, 0.52) | 0.50 (0.49, 0.51) |
| HuRL | 0.00 (0.00, 0.00) | 0.33 (0.32, 0.33) |
| PBRS | 0.00 (0.00, 0.00) | 0.22 (0.21, 0.22) |
| HEPO | **0.94 (0.85, 1.03)** | **0.73 (0.73, 0.74)** |

Table 1: IQM and probability of improvement (PI) over H-only (P(X > H-only)) with 95% confidence intervals across 12 heuristic reward functions (Section 4.2). H-only uses PPO with heuristic rewards. HEPO achieves higher normalized returns and a statistically significant PI greater than 0.5, indicating it significantly outperforms H-only.

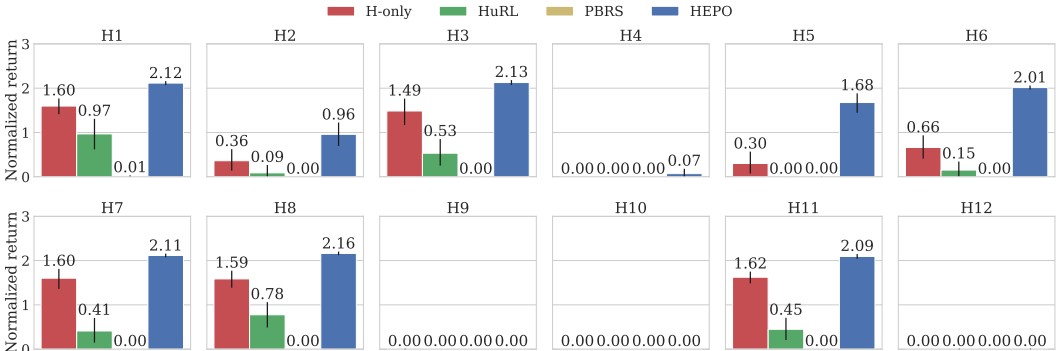

Figure 2: Normalized task return of PPO (H-only) and HEPO that are trained with heuristic reward function $H1$ to $H12$ designed by human subjects in the real world reward design condition. HEPO achieves higher task return than PPO (H-only) in 9 out of 12 tasks. This shows HEPO is robust to possibly ill-designed heuristic reward functions and can leverage them to improve performance.

ized returns below 1, while HEPO achieves returns greater than or close to 1. Since returns are normalized using the performance of the PPO policy trained with the well-designed heuristic reward function in ISAAC, a return below 1.0 indicates a performance drop for PPO (H-only) when using potentially ill-designed heuristic rewards. In contrast, HEPO can improve upon policies trained with carefully engineered heuristic reward functions, even when trained with possibly ill-designed heuristic reward functions.

**Qualitative observation:** We aim to understand why PPO's performance declines when trained with heuristic reward functions $H2$, $H5$, and $H6$. These functions are similar to the original heuristic reward in *FrankaCabinet*, but with different weights for each term. For example, in $H5$, the weight of action penalty is 1, whereas in the original heuristic reward function it is 7.5. This suggests that HEPO might handle poorly scaled heuristic reward terms better than PPO, which is sensitive to these weights. The heuristic reward functions $H12$ and $H9$ had an incorrect sign for the distance component, which caused the policy to be rewarded for moving away from the cabinet instead of toward it, making the learning task more challenging.

## 4.3 Ablation Studies

Expanding on the discussion of relation to relevant work EIPO [22] in Section 3.3, our goal is to examine the implementation choices of HEPO and illustrate the efficacy of each modification in this section. HEPO differs from EIPO primarily in two aspects: (1) the selection of a reference policy $\pi_{\text{ref}}$ in the constraint $J(\pi) \geq J(\pi_{\text{ref}})$, and (2) the strategy for utilizing policies to gather trajectories. Both studies are conducted on standard locomotion and manipulation tasks, such as *Ant*, *FrankaCabinet*, and *AllegroHand*. In addition, we provide further studies on the sensitivity to hyperparameters in Appendix B.2.

**Selection of reference policy in constraint:** HEPO and EIPO both enforce a performance improvement constraint $J(\pi) \geq J(\pi_{\text{ref}})$ during training. HEPO uses a heuristic policy $\pi_H$ as the reference ($\pi_{\text{ref}} = $ H-only), while EIPO uses a task-only policy ($\pi_{\text{ref}} = $ J-only). However, relying solely on policies trained with task rewards as references may not suffice for complex robotic tasks, as they often perform much worse than those trained with heuristic rewards. We compared the performance of HEPO with different reference policies in Figure 3a. The result shows that setting $\pi_{\text{ref}} = $ J-only (EIPO) improves the performance over the task-only policy *J-only* while notably degrading performance, sometimes even worse than *H-only*, suggesting it's insufficient for surpassing the heuristic policy.

**Strategy of collecting trajectories:** We use both the enhanced policy $\pi$ and the heuristic policy $\pi_H$ simultaneously to sample half of the environment's trajectories (referred to as *Joint*). Conversely, EIPO switches between $\pi$ and $\pi_H$ using a specified mechanism, where only one selected policy samples trajectories for updating both $\pi$ and $\pi_H$ within the same episode (referred to as *Alternating*). This study compares the performance of these two trajectory rollout methods. We modify HEPO to

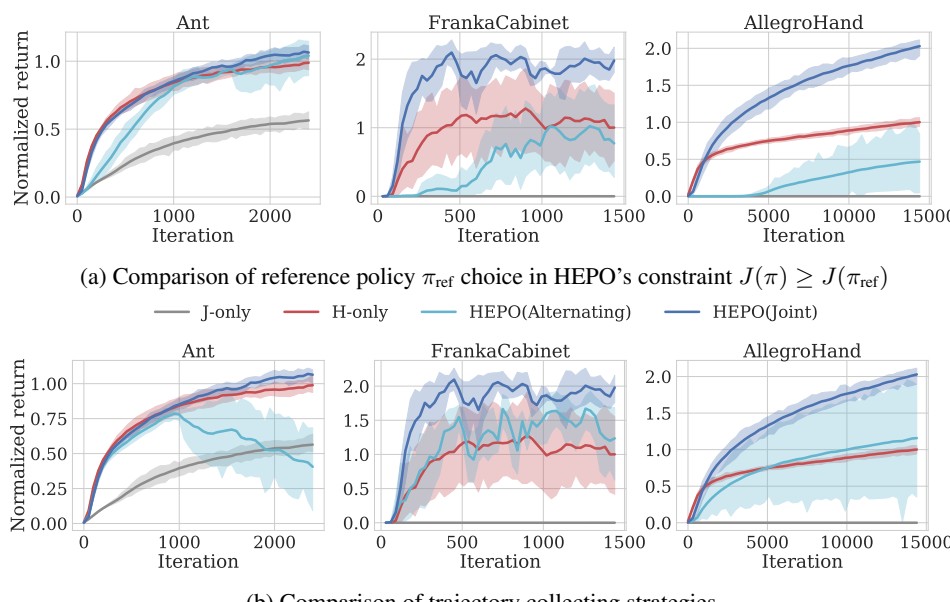

(a) Comparison of reference policy $\pi_{\text{ref}}$ choice in HEPO's constraint $J(\pi) \geq J(\pi_{\text{ref}})$

(b) Comparison of trajectory collecting strategies

Figure 3: **(a)** We show that using the policies trained with heuristic rewards (J-only) is better than using the policies trained with task rewards (J-only) when training HEPO. **(b)** *HEPO(Joint)* that collects trajectories using both policies leads to better performance than *HEPO(Alternating)* that alternates between two policies to collect trajectories. See Section 4.3 for details

gather trajectories using the *Alternating* strategy and present the results in Figure 3b. The findings indicate that *Alternating* results in a performance drop during mid-training and fails to match the performance of *HEPO(Joint)*. We hypothesize that this occurs because the batch of trajectories collected solely by one policy deviates significantly from those that another policy can generate (i.e., high off-policy error), leading to less effective PPO policy updates. In contrast, *Joint* samples trajectories using both policies, preventing the collected trajectories from deviating too much from each other.

# 5 Related Works

**Reward shaping:** Reward shaping has been a significant area, including potential-based reward shaping (PBRS) [2, 27], bilevel optimization approaches [28–30] on reward model learning, and heuristic-guided methods (HuRL) [10] that schedule heuristic rewards. Our method differs as it is a policy optimization method agnostic to the heuristic reward function and can be applied to those shaped or learned rewards.

**Constrained policy optimization:** Recent work like Extrinsic-Intrinsic Policy Optimization (EIPO) [22] proposes constrained optimization by tuning exploration bonuses to prevent exploiting them at the cost of task rewards. Extensions [31] balance imitating a teacher model and reward maximization. Our work differs in balancing human-designed heuristic rewards and task rewards, improving upon policies trained with engineered heuristic rewards. We also propose implementation enhancements over EIPO [22] (Section 4.3).

# 6 Discussion & Limitation

**HEPO for RL practitioners:** In this paper, we showed that HEPO is robust to the possibly ill-designed heuristic reward function in Section 4.2 and also exhibit high-probability improvement over PPO when training with heavily engineered heuristic rewards in robotic tasks in Section 4.1. Moving forward, when users need to integrate heuristic reward functions into RL algorithms, HEPO can potentially be a useful tool to reduce users' time on designing rewards since it can improve performance even with under-engineered heuristic rewards.

**Limitations:** While HEPO shows high-probability performance improvement over heuristic policies trained with well-designed heuristic reward, one limitation is that HEPO does not have a guarantee

to converge to the optimal policy theoretically. One future work can be incorporating the insight in recent theoretical advances on reward engineering [3] to make a convergence guarantee.

## Acknowledgements

We thank members of the Improbable AI Lab for helpful discussions and feedback. We are grateful to MIT Supercloud and the Lincoln Laboratory Supercomputing Center for providing HPC resources. This research was supported in part by Hyundai Motor Company, Quanta Computer Inc., an AWS MLRA research grant, ARO MURI under Grant Number W911NF-23-1-0277, DARPA Machine Common Sense Program, ARO MURI under Grant Number W911NF-21-1-0328, and ONR MURI under Grant Number N00014-22-1-2740. The views and conclusions contained in this document are those of the authors and should not be interpreted as representing the official policies, either expressed or implied, of the Army Research Office or the United States Air Force or the U.S. Government. The U.S. Government is authorized to reproduce and distribute reprints for Government purposes, notwithstanding any copyright notation herein.

## Author Contributions

- **Chi-Chang Lee:** Co-led the project, led the implementation of the proposed algorithms and baselines, and conducted the experiments.
- **Zhang-Wei Hong:** Co-led the project, led the writing of the paper, scaled up the experiment infrastructure, and conducted the experiments.
- **Pulkit Agrawal:** Played a key role in overseeing the project, editing the manuscript, and the presentation of the work.

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

# A  Implementation Details

## A.1  Full Derivation

We will detailedly describe the update of the enhanced policy ($\pi$ in Equation 7) and the heuristic policy ($\pi_H$ in Equation 8) at each iteration.

### A.1.1  Notations

- $V_r^\pi(s_t) := \mathbb{E}_{(s_t,a_t)\sim\pi}\Big[\sum_{t=0}^\infty \gamma^t r(s_t,a_t)|s_0 = s_t\Big]$

- $V_h^\pi(s_t) := \mathbb{E}_{(s_t,a_t)\sim\pi}\Big[\sum_{t=0}^\infty \gamma^t h(s_t,a_t)|s_0 = s_t\Big]$

- $A_r^\pi(s_t,a_t) := r(s_t,a_t) + V_r^\pi(s_{t+1}) - V_r^\pi(s_t)$

- $A_h^\pi(s_t,a_t) := h(s_t,a_t) + V_h^\pi(s_{t+1}) - V_h^\pi(s_t)$

- $B_{\text{HEPO}}$: the buffer to store samples collected by $\pi^i$

- $B_H$: the buffer to store samples collected $\pi_H^i$.

### A.1.2  Enhanced Policy $\pi$ Update

Given a $\alpha$ value, $\pi^{i+1}$ is derived using the arguments of the maxima in Equation 4, which can be re-written as follows:

$$
\begin{aligned}
\pi^{i+1} &= \arg\max_\pi \Big\{ J(\pi) + H(\pi) - \alpha\Big(J(\pi) - J(\pi_H^i)\Big)\Big\} \\
&= \arg\max_\pi \Big\{ (1+\alpha)J(\pi) + H(\pi)\Big\} \\
&= \arg\max_\pi \Big\{ \Big((1+\alpha)J(\pi) + H(\pi)\Big) - \frac{1}{2}\Big((1+\alpha)J(\pi^i) + H(\pi^i)\Big) \\
&\qquad - \frac{1}{2}\Big((1+\alpha)J(\pi_H^i) + H(\pi_H^i)\Big)\Big\} \\
&= \arg\max_\pi \Big\{ \frac{1}{2}\mathbb{E}_\pi\Big[(1+\alpha)A_r^{\pi^i}(s_t,a_t) + A_h^{\pi^i}(s_t,a_t)\Big] \\
&\qquad + \frac{1}{2}\mathbb{E}_\pi\Big[(1+\alpha)A_r^{\pi_H^i}(s_t,a_t) + A_h^{\pi_H^i}(s_t,a_t)\Big]\Big\} \\
&= \arg\max_\pi \Big\{ \frac{1}{2}\mathbb{E}_\pi\Big[U_\alpha^{\pi^i}(s_t,a_t)\Big] + \frac{1}{2}\mathbb{E}_\pi\Big[U_\alpha^{\pi_H^i}(s_t,a_t)\Big]\Big\} \\
&= \arg\max_\pi \Big\{ \mathbb{E}_\pi\Big[U_\alpha^{\pi^i}(s_t,a_t)\Big] + \mathbb{E}_\pi\Big[U_\alpha^{\pi_H^i}(s_t,a_t)\Big]\Big\}
\end{aligned}
\tag{12}
$$

where $U_\alpha^{\pi^i}$ and $U_\alpha^{\pi_H^i}$ are defined as follows:

$$
\begin{aligned}
U_\alpha^{\pi^i}(s_t,a_t) &:= (1+\alpha)A_r^{\pi^i}(s_t,a_t) + A_h^{\pi^i}(s_t,a_t) \\
U_\alpha^{\pi_H^i}(s_t,a_t) &:= (1+\alpha)A_r^{\pi_H^i}(s_t,a_t) + A_h^{\pi_H^i}(s_t,a_t)
\end{aligned}
\tag{13}
$$

To efficiently achieve the update process in Equation 12, we aim to utilize previously collected trajectories for optimization, outlined in Equation 7. Here, we refer to [15], using those previously collected trajectories to form a lower bound surrogate objectives, $\hat{J}_\alpha^{\pi^i}(\pi)$ and $\hat{J}_\alpha^{\pi_H^i}(\pi)$, as alternatives

of $\mathbb{E}_\pi[U_\alpha^{\pi^i}(s_t, a_t)]$ and $\mathbb{E}_\pi[U_\alpha^{\pi_H^i}(s_t, a_t)]$ to derive $\pi^{i+1}$:

$$
\hat{J}_{\text{HEPO}}^{\pi^i}(\pi) := \frac{1}{|B_{\text{HEPO}}|} \sum_{(s_t, a_t) \in B_{\text{HEPO}}} \Big[ \sum_{t=0}^{\infty} \gamma^t \min \Big\{ \frac{\pi(a_t|s_t)}{\pi^i(a_t|s_t)} U_\alpha^{\pi^i}(s_t, a_t),
$$
$$
\text{clip}\left( \frac{\pi(a_t|s_t)}{\pi^i(a_t|s_t)}, 1-\epsilon, 1+\epsilon \right) U_\alpha^{\pi^i}(s_t, a_t) \Big\} \Big]
$$
$$
\hat{J}_{\text{HEPO}}^{\pi_H^i}(\pi) := \frac{1}{|B_H|} \sum_{(s_t, a_t) \in B_H} \Big[ \sum_{t=0}^{\infty} \gamma^t \min \Big\{ \frac{\pi(a_t|s_t)}{\pi_H^i(a_t|s_t)} U_\alpha^{\pi_H^i}(s_t, a_t),
$$
$$
\text{clip}\left( \frac{\pi(a_t|s_t)}{\pi_H^i(a_t|s_t)}, 1-\epsilon, 1+\epsilon \right) U_\alpha^{\pi_H^i}(s_t, a_t) \Big\} \Big],
$$
(14)

where $\mathbb{E}_\pi[U_\alpha^{\pi^i}(s_t, a_t)] \geq \hat{J}_{\text{HEPO}}^{\pi^i}(\pi)$ and $\mathbb{E}_\pi[U_\alpha^{\pi_H^i}(s_t, a_t)] \geq \hat{J}_{\text{HEPO}}^{\pi_H^i}(\pi)$ always hold; $\epsilon \in [0, 1]$ denotes a threshold. Intuitively, this clipped objective (Eq. 14) penalizes the policy $\pi$ that behaves differently from $\pi^i$ or $\pi_H^i$ because overly large or small the action probability ratios between two policies are clipped.

### A.1.3 Heuristic Policy $\pi_H$ Update

$\pi_H^{i+1}$ is derived using the arguments of the maxima of $H(\pi)$, which can be re-written as follows:

$$
\pi_H^{i+1} = \arg\max_\pi \Big\{ H(\pi) \Big\}
$$
$$
= \arg\max_\pi \Big\{ H(\pi) - \frac{1}{2} H(\pi^i) - \frac{1}{2} H(\pi_H^i) \Big\}
$$
$$
= \arg\max_\pi \Big\{ \frac{1}{2} \mathbb{E}_\pi \Big[ A_h^{\pi^i}(s_t, a_t) \Big] + \frac{1}{2} \mathbb{E}_\pi \Big[ A_h^{\pi_H^i}(s_t, a_t) \Big] \Big\}
$$
$$
= \arg\max_\pi \Big\{ \mathbb{E}_\pi \Big[ A_h^{\pi^i}(s_t, a_t) \Big] + \mathbb{E}_\pi \Big[ A_h^{\pi_H^i}(s_t, a_t) \Big] \Big\}
$$
(15)

Similarly, we again rely on the approximation from [15] to derive a lower bound surrogate objective for both $\mathbb{E}_\pi[A_h^{\pi^i}(s_t, a_t)]$ and $\mathbb{E}_\pi[A_h^{\pi_H^i}(s_t, a_t)]$ as follows:

$$
\hat{H}^{\pi^i}(\pi) := \frac{1}{|B_{\text{HEPO}}|} \sum_{(s_t, a_t) \in B_{\text{HEPO}}} \Big[ \sum_{t=0}^{\infty} \gamma^t \min \Big\{ \frac{\pi(a_t|s_t)}{\pi^i(a_t|s_t)} A_h^{\pi^i}(s_t, a_t),
$$
$$
\text{clip}\left( \frac{\pi(a_t|s_t)}{\pi^i(a_t|s_t)}, 1-\epsilon, 1+\epsilon \right) A_h^{\pi^i}(s_t, a_t) \Big\} \Big],
$$
(16)
$$
\hat{H}^{\pi_H^i}(\pi) := \frac{1}{|B_H|} \sum_{(s_t, a_t) \in B_H} \Big[ \sum_{t=0}^{\infty} \gamma^t \min \Big\{ \frac{\pi(a_t|s_t)}{\pi_H^i(a_t|s_t)} A_h^{\pi_H^i}(s_t, a_t),
$$
$$
\text{clip}\left( \frac{\pi(a_t|s_t)}{\pi_H^i(a_t|s_t)}, 1-\epsilon, 1+\epsilon \right) A_h^{\pi_H^i}(s_t, a_t) \Big\} \Big]
$$
(17)

where $\mathbb{E}_\pi[A_h^{\pi^i}(s_t, a_t)] \geq \hat{H}^{\pi^i}(\pi)$ and $\mathbb{E}_\pi[A_h^{\pi_H^i}(s_t, a_t)] \geq \hat{H}^{\pi_H^i}(\pi)$ always hold. Different from vanilla heuristic training, instead of solely collecting trajectories from $\pi_H^i$, we collect trajectories from both $\pi$ and $\pi_H$ to enrich sample efficiency.

### A.2 Implementation Tricks

### A.2.1 Sample Sharing for Value Function Update

In practice, obtaining real value functions for training is not feasible. We estimate the value function using collected trajectories, but this approach tends to fail because the value function becomes biased toward the policy responsible for trajectory collection.

To prevent error information from the estimated value function interfering with the training procedure, we share the trajectory samples within the $B_{\text{HEPO}}$ and $B_H$ buffers to update our value functions:

$$V_r^{\pi^{i+1}} \leftarrow \arg\min_V \left\{ \sum_{\left(s_t, r(s_t,a_t), s_{t+1}\right) \in B_{\text{HEPO}} \cup B_H} \frac{|r(s_t,a_t) + \gamma V_r^{\pi^i}(s_{t+1}) - V(s_t)|^2}{|B_{\text{HEPO}}| + |B_H|} \right\} \quad (18)$$

$$V_h^{\pi^{i+1}} \leftarrow \arg\min_V \left\{ \sum_{\left(s_t, h(s_t,a_t), s_{t+1}\right) \in B_{\text{HEPO}} \cup B_H} \frac{|h(s_t,a_t) + \gamma V_h^{\pi^i}(s_{t+1}) - V(s_t)|^2}{|B_{\text{HEPO}}| + |B_H|} \right\} \quad (19)$$

$$V_r^{\pi_H^{i+1}} \leftarrow \arg\min_V \left\{ \sum_{\left(s_t, r(s_t,a_t), s_{t+1}\right) \in B_{\text{HEPO}} \cup B_H} \frac{|r(s_t,a_t) + \gamma V_r^{\pi_H^i}(s_{t+1}) - V(s_t)|^2}{|B_{\text{HEPO}}| + |B_H|} \right\} \quad (20)$$

$$V_h^{\pi_H^{i+1}} \leftarrow \arg\min_V \left\{ \sum_{\left(s_t, h(s_t,a_t), s_{t+1}\right) \in B_{\text{HEPO}} \cup B_H} \frac{|h(s_t,a_t) + \gamma V_h^{\pi_H^i}(s_{t+1}) - V(s_t)|^2}{|B_{\text{HEPO}}| + |B_H|} \right\} \quad (21)$$

### A.2.2   Smoothing Lagrangian Multiplier $\alpha$ Update

The Lagrangian multiplier $\alpha$ determines the desired constraint information during training. However, in practice the gradient $\alpha$ tends to become explosive. To stabilize the $\alpha$ update procedure, we accumulate previous gradients and adopt the Adam optimizer [32] as follows:

$$g(\alpha) \leftarrow \text{med}\left\{ \frac{1}{|B_{\text{HEPO}}|} \sum_{(s_t,a_t) \in B_{\text{HEPO}}} \left[ A_r^{\pi_H^i}(s_t,a_t) \right] - \frac{1}{|B_H|} \sum_{(s_t,a_t) \in B_H} \left[ A_r^{\pi^i}(s_t,a_t) \right] \right\}_{i-K}^{i}$$

$$\alpha \leftarrow \text{AdamOpt}\left[ g(\alpha) \right] \quad (22)$$

where $K$ is the number of previous $K$ advantage expectation records that we take into account. To smooth the current $\alpha$ gradient for each update, we calculate the median of the previous $K$ records. In our experiments, we assigned $K$ a value of $8$.

### A.3   Overall Workflow

---

**Algorithm 2** Detailed Heuristic-Enhanced Policy Optimization (HEPO)

---

1: Initialize policies $(\pi^1, \pi_H^1)$ and values $(V_r^{\pi^1}, V_h^{\pi^1}, V_r^{\pi_H^1}, V_h^{\pi_H^1})$
2: **for** $i = 1 \cdots$ **do**                                    $\triangleright i$ denotes iteration index
3:     # ROLLOUT STAGE
4:     Collect trajectory buffers $(B_{\text{HEPO}}, B_H)$ using $(\pi^i, \pi_H^i)$
5:     Compute $\left(A_r^{\pi^i}(s_t,a_t), A_h^{\pi^i}(s_t,a_t)\right)$ via GAE with $\left(V_r^{\pi^i}, V_h^{\pi^i}\right) \forall (s_t,a_t) \in B_{\text{HEPO}}$
6:     Compute $\left(A_r^{\pi_H^i}(s_t,a_t), A_h^{\pi_H^i}(s_t,a_t)\right)$ via GAE with $\left(V_r^{\pi_H^i}, V_h^{\pi_H^i}\right) \forall (s_t,a_t) \in B_H$
7:     Compute $\left(\hat{J}_{\text{HEPO}}^{\pi^i}, \hat{J}_{\text{HEPO}}^{\pi_H^i}\right)$ based on Equation 14
8:     Compute $\left(\hat{H}^{\pi^i}, \hat{H}^{\pi_H^i}\right)$ based on Equation 16
9:
10:     # UPDATE STAGE
11:     $\pi^{i+1} \leftarrow \arg\max_\pi \left\{ \hat{J}_{\text{HEPO}}^{\pi^i}(\pi) + \hat{J}_{\text{HEPO}}^{\pi_H^i}(\pi) \right\}$
12:     $\pi_H^{i+1} \leftarrow \arg\max_\pi \left\{ \hat{H}^{\pi^i}(\pi) + \hat{H}^{\pi_H^i}(\pi) \right\}$
13:     Update $(V_r^{\pi^i}, V_h^{\pi^i}, V_r^{\pi_H^i}, V_h^{\pi_H^i})$ based on Equation 18
14:     Update $\alpha$ based on Equation 22
15: **end for**

---

### A.4 Training details

Following the PPO framework [15], our experiments are based on a continuous action actor-critic algorithm implemented in `rl_games` [25], using Generalized Advantage Estimation (GAE) [21] to compute advantages for policy optimization. For PPO, we employed the same policy network and value network architecture, and the same hyperparameters used in `IsaacGymEnvs` [19]. We also include our source code in the supplementary material. In HEPO, we use two policies for optimization, with each policy maintaining the same model configurations as those used in PPO. Below we introduce HEPO-specific hyperparameters used in our experiments in Section 4.1. The hyperparameters for updating the Lagrangian multiplier $\alpha$ in HEPO are listed as follows:

Table 2: HEPO Hyperparameters

| Name | Value |
|---|---|
| Initial $\alpha$ | 0.0 |
| Step size $\eta$ of $\alpha$ (learning rate) | 0.01 |
| Clipping range of $\delta\alpha$ $(-\epsilon_\alpha, \epsilon_\alpha)$ | 1.0 |
| Range of the $\alpha$ value | $[0, \infty)$ |

For baselines, we search for hyperparamters $\lambda$ for $J + H$ in *Ant*, *FrankaCabinet*, and *AllegroHand*, as shown in Section B.2. We set $\lambda = 1$ for all the experiments because it shows better performance on the three chosen environments. For HuRL [10], we follow the scheduling setting provided in their paper.

# B Supplementary Experimental Results

## B.1 All Learning Curves on the task objective $J$

We present all the learning curves in Figure 4.

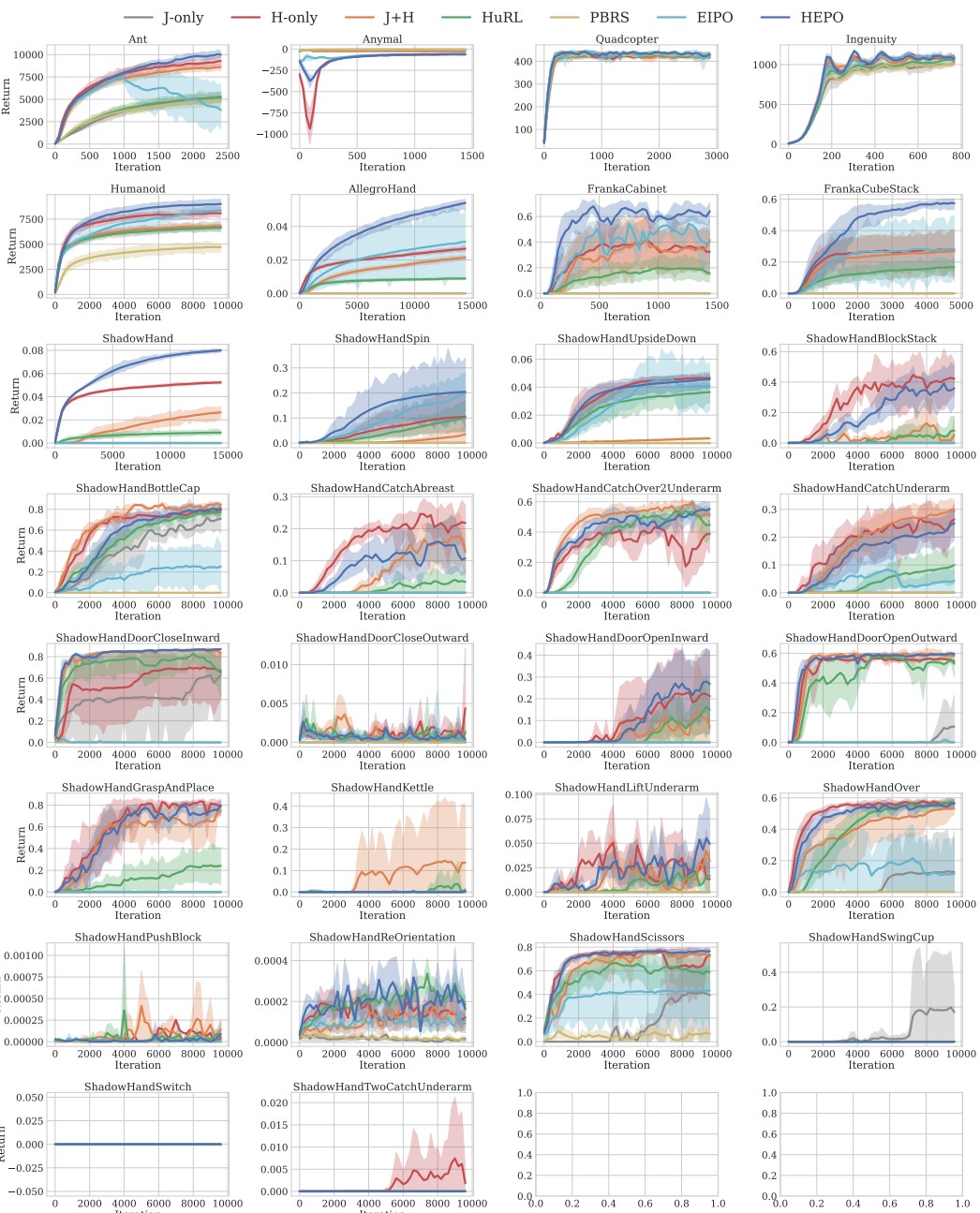

Figure 4: All learning curves in Section 4.1

## B.2 Sensitivity to hyperparameters

In this section, we aim to verify HEPO's sensitivity to two main types of hyperparameters: (1) the weight of the heuristic reward in optimization (denoted as $\lambda$) and (2) the learning rate for updating $\alpha$.

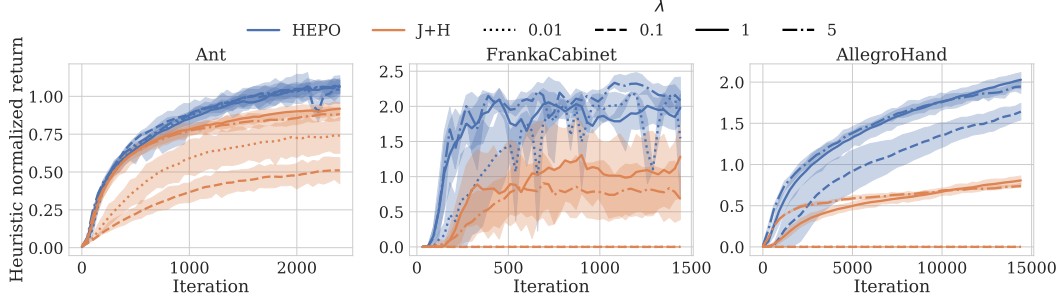

Figure 5: Sensitivity to $\lambda$

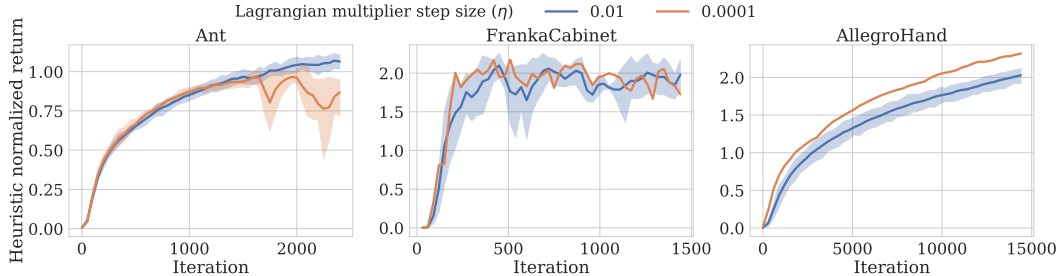

Figure 6: Sensitivity to alpha learning rate

Similar to Section 4.3, we conducted our experiments on the *Ant*, *FrankaCabinet*, and *AllegroHand* tasks.

### B.2.1 Sensitivity to the $\lambda$ Value

Both **HEPO** and **J+H** can set a scaling coefficient to weight the heuristic reward in optimization, such that the objective becomes $J(\pi) + \lambda H(\pi)$. This scaling coefficient can be used to balance both objectives. In this study, we compare **HEPO** and **J+H** on their performance sensitivity to the choice of $\lambda$, exhaustively training both **HEPO** and **J+H** with varying $\lambda$ values. Note that though the formulation of HEPO does not depend on $\lambda$, one can still set a $\lambda$ coefficient to scale the heuristic reward in HEPO. In our experiments, we did not optimize $\lambda$ for HEPO but for the baselines trained with both rewards (J+H). In Figure 5, we found that **J+H** is sensitive to $\lambda$ in all selected tasks, while **HEPO** performs well across a wide range of $\lambda$ values. This indicates that **HEPO** is robust to the choice of $\lambda$.

### B.2.2 Sensitivity to the Learning Rate for Updating $\alpha$

HEPO's robustness relies on the $\alpha$ update, as it reflects the necessary constraint information at each iteration. Similar to Appendix B.2.1, setting different initial values of $\alpha$ is equivalent to using different $\lambda$ values for our estimation, since both can be rewritten as the ratio between $H(\pi)$ and $J(\pi)$. Both of these parameters indicate the necessary constraint information for conducting multi-objective optimization. In this study, we aim to verify whether **HEPO** can yield comparable improvement gaps under different initial values of $\alpha$, thus providing a more robust optimization procedure.

As shown in Figure 6, we observe that **HEPO** is also robust to the choice of $\alpha$'s initial values, similar to the results in Figure 5.

### B.3 Additional results

**Generality of HEPO:** We also demonstrated HEPO can be implemented over the other RL algorithms in addition to PPO. We integrated HEPO into HuRL's SAC codebase. Despite HuRL using tuned hyperparameters as reported in its paper [10], HEPO outperformed SAC and matched HuRL on the

most challenging task in Figure 7b using the same hyperparameters from Section 4 of our manuscript, showing the generality of HEPO on different RL algorithms.

**Better than EIPO on the most challenging task:** Comparing HEPO and EIPO on the most challenging task, Montezuma's Revenge, reported in EIPO's paper [22] using RND exploration bonuses [23] (as used in the EIPO paper), Figure 7a shows that HEPO performs better than EIPO. Also, HEPO matched PPO trained with RND bonuses at convergence (2 billion frames) reported in [23] using only 20% of the training data, demonstrating drastically improved sample efficiency.

**How quality of heuristic reward functions impact HEPO?** We believe that Figure 2 reveals the relationship between HEPO's performance and the quality of the heuristic reward. The policy trained with only heuristic rewards (H-only) represents both the asymptotic performance of in HEPO and the quality of the heuristic itself. We found a positive correlation (Pearson coefficient of 0.9) between the average performances of H-only and HEPO in Figure 2 results, suggesting that better heuristics lead to improved HEPO performance. Figure 7c provides more details.

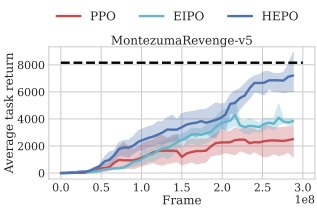
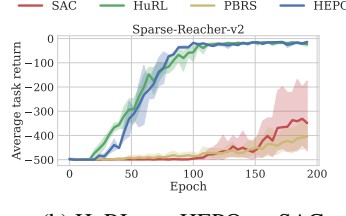
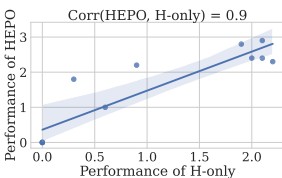

(a) EIPO v.s. HEPO on PPO    (b) HuRL v.s. HEPO on SAC    (c) Correlation between the performance of HEPO and H-only.

Figure 7: **(a)** Comparison of HEPO and EIPO [22] on the most challenging Atari task, `Montezuma's Revenge`, shown in the EIPO paper [22]. Both are implemented on top of EIPO's PPO codebase using RND exploration bonuses [23] as heuristic rewards $H$, as suggested in [22]. HEPO outperforms EIPO, achieving the performance (denoted as dashed line) similar to PPO trained with RND at convergence (2 billion frames) reported in [23] in five times fewer frames. **(b)** HEPO matches HuRL's performance on the most challenging `Sparse-Reacher` task using HuRL's SAC codebase [10], despite HuRL being tuned for this task and HEPO using the same hyperparameters from our Section 4. This also highlights HEPO's generality in different RL algorithms. **(c)** HEPO's performance is positively correlated with that of the heuristic policy trained with heuristic rewards only (H-only), suggesting that HEPO's effectiveness will improve as the quality of heuristic rewards increases.

## C  Environment Details

As depicted in Section 4, we conducted our experiments based on the Isaac Gym (**ISAAC**) simulator [19] and the Bi-DexHands (**BI-DEX**) benchmark [24]. The selected task classes in ISAAC can be partitioned into 4 groups - Locomotion Tracking (***Anymal***), Locomotion Progressing (***Ant*** and ***Humanoid***), Helicopter Progressing (***Ingenuity*** and ***Quadcopter***), and Manipulation Tasks (***FrankaCabinet***, ***FrankaCubeStack***, ***ShadowHand***, and ***AllegroHand***). In addition, BI-DEX provides dual dexterous hand manipulation tasks through ISAAC, reaching human-level sophistication of hand dexterity and bimanual coordination. Their tasks include ***ShadowHandOver***, ***ShadowHandCatchUnderarm***, ***ShadowHandCatchOver2Underarm***, ***ShadowHandCatchAbreast***, ***ShadowHandTwoCatchUnderarm***, ***ShadowHandLiftUnderarm***, ***ShadowHandDoorOpenInward***, ***ShadowHandDoorOpenOutward***, ***ShadowHandDoorCloseInward***, ***ShadowHandDoorCloseOutward***, ***ShadowHandSpin***, ***ShadowHandUpsideDown***, ***ShadowHandBlockStack***, ***ShadowHandBottleCap***, ***ShadowHandGraspAndPlace***, ***ShadowHandKettle***, ***ShadowHandPen***, ***ShadowHandPushBlock***, ***ShadowHandReOrientation***, ***ShadowHandScissors***, ***ShadowHandSwingCup***.

For Locomotion Tracking, our emphasis lies in assessing the precision of velocities in linear and angular motions, ensuring that the robot responds closely to the assigned values. To this end, we define tracking errors as our task rewards. For Locomotion Progressing and Helicopter Progressing, our emphasis lies in evaluating the progress made by the robots in reaching the assigned destination from a given start point. To this end, we define movement progress as our task rewards. For Manipulation tasks and all tasks within the BI-DEX benchmark, our emphasis lies in whether and how quickly the robotic hands can successfully complete the assigned missions, reaching the desired goal states. To

this end, we define task rewards using a binary label, assigning a value of 1 to indicate successful attainment of the goal state and 0 otherwise.

The following are our reward function definitions, which include heuristic and task reward terms in Python style:

Isaac Gym - Locomotion Tracking Task: *Anymal*

```python
def compute_anymal_reward(
    root_states,
    commands,
    torques,
    contact_forces,
    knee_indices,
    episode_lengths,
    rew_scales,
    base_index,
    max_episode_length
):
    # (reward, reset, feet_in air, feet_air_time, episode sums)
    # type: (Tensor, Tensor, Tensor, Tensor, Tensor, Tensor, Dict[str, float], int, int) -> Tuple[Tensor,
    ↪   Tensor, Tensor]

    # prepare quantities (TODO: return from obs ?)
    base_quat = root_states[:, 3:7]
    base_lin_vel = quat_rotate_inverse(base_quat, root_states[:, 7:10])
    base_ang_vel = quat_rotate_inverse(base_quat, root_states[:, 10:13])

    # velocity tracking reward
    lin_vel_error = torch.sum(torch.square(commands[:, :2] - base_lin_vel[:, :2]), dim=1)
    ang_vel_error = torch.square(commands[:, 2] - base_ang_vel[:, 2])
    rew_lin_vel_xy = torch.exp(-lin_vel_error/0.25) * rew_scales["lin_vel_xy"]
    rew_ang_vel_z = torch.exp(-ang_vel_error/0.25) * rew_scales["ang_vel_z"]

    # torque penalty
    rew_torque = torch.sum(torch.square(torques), dim=1) * rew_scales["torque"]

    total_reward = rew_lin_vel_xy + rew_ang_vel_z + rew_torque
    total_reward = torch.clip(total_reward, 0., None)
    tracking_reward = -(lin_vel_error + ang_vel_error)

    # reset agents
    reset = torch.norm(contact_forces[:, base_index, :], dim=1) > 1.
    reset = reset | torch.any(torch.norm(contact_forces[:, knee_indices, :], dim=2) > 1., dim=1)
    time_out = episode_lengths >= max_episode_length - 1  # no terminal reward for time-outs
    reset = reset | time_out
    heuristic_reward, task_reward = total_reward.detach(), tracking_reward
    return heuristic_reward, task_reward, reset
```

Isaac Gym - Locomotion Progressing Task: *Ant*

```python
def compute_ant_reward(
    obs_buf,
    reset_buf,
    progress_buf,
    actions,
    up_weight,
    heading_weight,
    potentials,
    prev_potentials,
    actions_cost_scale,
    energy_cost_scale,
    joints_at_limit_cost_scale,
    termination_height,
    death_cost,
    max_episode_length
):
    # type: (Tensor, Tensor, Tensor, Tensor, float, float, Tensor, Tensor, float, float, float, float,
    ↪  float, float) -> Tuple[Tensor, Tensor, Tensor]

    # reward from direction headed
    heading_weight_tensor = torch.ones_like(obs_buf[:, 11]) * heading_weight
    heading_reward = torch.where(obs_buf[:, 11] > 0.8, heading_weight_tensor, heading_weight * obs_buf[:,
    ↪  11] / 0.8)

    # aligning up axis of ant and environment
    up_reward = torch.zeros_like(heading_reward)
    up_reward = torch.where(obs_buf[:, 10] > 0.93, up_reward + up_weight, up_reward)

    # energy penalty for movement
    actions_cost = torch.sum(actions ** 2, dim=-1)
    electricity_cost = torch.sum(torch.abs(actions * obs_buf[:, 20:28]), dim=-1)
    dof_at_limit_cost = torch.sum(obs_buf[:, 12:20] > 0.99, dim=-1)

    # reward for duration of staying alive
    alive_reward = torch.ones_like(potentials) * 0.5
    progress_reward = potentials - prev_potentials

    total_reward = progress_reward + alive_reward + up_reward + heading_reward - \
        actions_cost_scale * actions_cost - energy_cost_scale * electricity_cost - dof_at_limit_cost *
        ↪  joints_at_limit_cost_scale

    # adjust reward for fallen agents
    total_reward = torch.where(obs_buf[:, 0] < termination_height, torch.ones_like(total_reward) *
    ↪  death_cost, total_reward)

    # reset agents
    reset = torch.where(obs_buf[:, 0] < termination_height, torch.ones_like(reset_buf), reset_buf)
    reset = torch.where(progress_buf >= max_episode_length - 1, torch.ones_like(reset_buf), reset)
    heuristic_reward, task_reward = total_reward, progress_reward
    return heuristic_reward, task_reward, reset
```

Isaac Gym - Locomotion Progressing Task: *Humanoid*

```python
def compute_humanoid_reward(
    obs_buf,
    reset_buf,
    progress_buf,
    actions,
    up_weight,
    heading_weight,
    potentials,
    prev_potentials,
    actions_cost_scale,
    energy_cost_scale,
    joints_at_limit_cost_scale,
    max_motor_effort,
    motor_efforts,
    termination_height,
    death_cost,
    max_episode_length
):
    # type: (Tensor, Tensor, Tensor, Tensor, float, float, Tensor, Tensor, float, float, float, float,
    ↪  Tensor, float, float, float) -> Tuple[Tensor, Tensor, Tensor]

    # reward from the direction headed
    heading_weight_tensor = torch.ones_like(obs_buf[:, 11]) * heading_weight
    heading_reward = torch.where(obs_buf[:, 11] > 0.8, heading_weight_tensor, heading_weight * obs_buf[:,
    ↪  11] / 0.8)

    # reward for being upright
    up_reward = torch.zeros_like(heading_reward)
    up_reward = torch.where(obs_buf[:, 10] > 0.93, up_reward + up_weight, up_reward)

    actions_cost = torch.sum(actions ** 2, dim=-1)

    # energy cost reward
    motor_effort_ratio = motor_efforts / max_motor_effort
    scaled_cost = joints_at_limit_cost_scale * (torch.abs(obs_buf[:, 12:33]) - 0.98) / 0.02
    dof_at_limit_cost = torch.sum((torch.abs(obs_buf[:, 12:33]) > 0.98) * scaled_cost *
    ↪  motor_effort_ratio.unsqueeze(0), dim=-1)

    electricity_cost = torch.sum(torch.abs(actions * obs_buf[:, 33:54]) *
    ↪  motor_effort_ratio.unsqueeze(0), dim=-1)

    # reward for duration of being alive
    alive_reward = torch.ones_like(potentials) * 2.0
    progress_reward = potentials - prev_potentials

    total_reward = progress_reward + alive_reward + up_reward + heading_reward - \
        actions_cost_scale * actions_cost - energy_cost_scale * electricity_cost - dof_at_limit_cost

    # adjust reward for fallen agents
    total_reward = torch.where(obs_buf[:, 0] < termination_height, torch.ones_like(total_reward) *
    ↪  death_cost, total_reward)

    # reset agents
    reset = torch.where(obs_buf[:, 0] < termination_height, torch.ones_like(reset_buf), reset_buf)
    reset = torch.where(progress_buf >= max_episode_length - 1, torch.ones_like(reset_buf), reset)
    heuristic_reward, task_reward = total_reward, progress_reward
    return heuristic_reward, task_reward, reset
```

## Isaac Gym - Helicopter Progressing Task: *Ingenuity*

```python
def compute_ingenuity_reward(root_positions, target_root_positions, root_quats, root_linvels,
↪    root_angvels, reset_buf, progress_buf, max_episode_length):
    # type: (Tensor, Tensor, Tensor, Tensor, Tensor, Tensor, Tensor, float) -> Tuple[Tensor, Tensor,
    ↪    Tensor]

    # distance to target
    target_dist = torch.sqrt(torch.square(target_root_positions - root_positions).sum(-1))
    pos_reward = 1.0 / (1.0 + target_dist * target_dist)

    # uprightness
    ups = quat_axis(root_quats, 2)
    tiltage = torch.abs(1 - ups[..., 2])
    up_reward = 5.0 / (1.0 + tiltage * tiltage)

    # spinning
    spinnage = torch.abs(root_angvels[..., 2])
    spinnage_reward = 1.0 / (1.0 + spinnage * spinnage)

    # combined reward
    # uprigness and spinning only matter when close to the target
    reward = pos_reward + pos_reward * (up_reward + spinnage_reward)

    # resets due to misbehavior
    ones = torch.ones_like(reset_buf)
    die = torch.zeros_like(reset_buf)
    die = torch.where(target_dist > 8.0, ones, die)
    die = torch.where(root_positions[..., 2] < 0.5, ones, die)

    # resets due to episode length
    reset = torch.where(progress_buf >= max_episode_length - 1, ones, die)
    heuristic_reward, task_reward = reward, pos_reward
    return heuristic_reward, task_reward, reset
```

## Isaac Gym - Helicopter Progressing Task: *Quadcopter*

```python
def compute_quadcopter_reward(root_positions, root_quats, root_linvels, root_angvels, reset_buf,
↪    progress_buf, max_episode_length):
    # type: (Tensor, Tensor, Tensor, Tensor, Tensor, Tensor, float) -> Tuple[Tensor, Tensor, Tensor]

    # distance to target
    target_dist = torch.sqrt(root_positions[..., 0] * root_positions[..., 0] +
                             root_positions[..., 1] * root_positions[..., 1] +
                             (1 - root_positions[..., 2]) * (1 - root_positions[..., 2]))
    pos_reward = 1.0 / (1.0 + target_dist * target_dist)

    # uprightness
    ups = quat_axis(root_quats, 2)
    tiltage = torch.abs(1 - ups[..., 2])
    up_reward = 1.0 / (1.0 + tiltage * tiltage)

    # spinning
    spinnage = torch.abs(root_angvels[..., 2])
    spinnage_reward = 1.0 / (1.0 + spinnage * spinnage)

    # combined reward
    # uprigness and spinning only matter when close to the target
    reward = pos_reward + pos_reward * (up_reward + spinnage_reward)

    # resets due to misbehavior
    ones = torch.ones_like(reset_buf)
    die = torch.zeros_like(reset_buf)
    die = torch.where(target_dist > 3.0, ones, die)
    die = torch.where(root_positions[..., 2] < 0.3, ones, die)

    # resets due to episode length
    reset = torch.where(progress_buf >= max_episode_length - 1, ones, die)
    heuristic_reward, task_reward = reward, pos_reward
    return heuristic_reward, task_reward, reset
```

## Isaac Gym - Manipulation Task: *FrankaCabinet*

```python
def compute_franka_reward(
    reset_buf, progress_buf, reset_goal_buf, successes, consecutive_successes, actions, cabinet_dof_pos,
    franka_grasp_pos, drawer_grasp_pos, franka_grasp_rot, drawer_grasp_rot,
    franka_lfinger_pos, franka_rfinger_pos,
    gripper_forward_axis, drawer_inward_axis, gripper_up_axis, drawer_up_axis,
    num_envs, dist_reward_scale, rot_reward_scale, around_handle_reward_scale, open_reward_scale,
    finger_dist_reward_scale, action_penalty_scale, distX_offset, max_episode_length
):
    # type: (Tensor, Tensor, Tensor, Tensor, Tensor, Tensor, Tensor, Tensor, Tensor, Tensor, Tensor,
    ↪   Tensor, Tensor, Tensor, Tensor, Tensor, Tensor, int, float, float, float, float, float, float,
    ↪   float, float) -> Tuple[Tensor, Tensor, Tensor, Tensor]

    # distance from hand to the drawer
    d = torch.norm(franka_grasp_pos - drawer_grasp_pos, p=2, dim=-1)
    dist_reward = 1.0 / (1.0 + d ** 2)
    dist_reward *= dist_reward
    dist_reward = torch.where(d <= 0.02, dist_reward * 2, dist_reward)

    axis1 = tf_vector(franka_grasp_rot, gripper_forward_axis)
    axis2 = tf_vector(drawer_grasp_rot, drawer_inward_axis)
    axis3 = tf_vector(franka_grasp_rot, gripper_up_axis)
    axis4 = tf_vector(drawer_grasp_rot, drawer_up_axis)

    dot1 = torch.bmm(axis1.view(num_envs, 1, 3), axis2.view(num_envs, 3, 1)).squeeze(-1).squeeze(-1)  #
    ↪   alignment of forward axis for gripper
    dot2 = torch.bmm(axis3.view(num_envs, 1, 3), axis4.view(num_envs, 3, 1)).squeeze(-1).squeeze(-1)  #
    ↪   alignment of up axis for gripper
    # reward for matching the orientation of the hand to the drawer (fingers wrapped)
    rot_reward = 0.5 * (torch.sign(dot1) * dot1 ** 2 + torch.sign(dot2) * dot2 ** 2)

    # bonus if left finger is above the drawer handle and right below
    around_handle_reward = torch.zeros_like(rot_reward)
    around_handle_reward = torch.where(franka_lfinger_pos[:, 2] > drawer_grasp_pos[:, 2],
                                       torch.where(franka_rfinger_pos[:, 2] < drawer_grasp_pos[:, 2],
                                                   around_handle_reward + 0.5, around_handle_reward),
                                                   ↪   around_handle_reward)
    # reward for distance of each finger from the drawer
    finger_dist_reward = torch.zeros_like(rot_reward)
    lfinger_dist = torch.abs(franka_lfinger_pos[:, 2] - drawer_grasp_pos[:, 2])
    rfinger_dist = torch.abs(franka_rfinger_pos[:, 2] - drawer_grasp_pos[:, 2])
    finger_dist_reward = torch.where(franka_lfinger_pos[:, 2] > drawer_grasp_pos[:, 2],
                                     torch.where(franka_rfinger_pos[:, 2] < drawer_grasp_pos[:, 2],
                                                 (0.04 - lfinger_dist) + (0.04 - rfinger_dist),
                                                 ↪   finger_dist_reward), finger_dist_reward)

    # regularization on the actions (summed for each environment)
    action_penalty = torch.sum(actions ** 2, dim=-1)

    # how far the cabinet has been opened out
    open_reward = cabinet_dof_pos[:, 3] * around_handle_reward + cabinet_dof_pos[:, 3]  #
    ↪   drawer_top_joint

    rewards = dist_reward_scale * dist_reward + rot_reward_scale * rot_reward \
        + around_handle_reward_scale * around_handle_reward + open_reward_scale * open_reward \
        + finger_dist_reward_scale * finger_dist_reward - action_penalty_scale * action_penalty

    # bonus for opening drawer properly
    rewards = torch.where(cabinet_dof_pos[:, 3] > 0.01, rewards + 0.5, rewards)
    rewards = torch.where(cabinet_dof_pos[:, 3] > 0.2, rewards + around_handle_reward, rewards)
    rewards = torch.where(cabinet_dof_pos[:, 3] > 0.39, rewards + (2.0 * around_handle_reward), rewards)

    # prevent bad style in opening drawer
    rewards = torch.where(franka_lfinger_pos[:, 0] < drawer_grasp_pos[:, 0] - distX_offset,
                          torch.ones_like(rewards) * -1, rewards)
    rewards = torch.where(franka_rfinger_pos[:, 0] < drawer_grasp_pos[:, 0] - distX_offset,
                          torch.ones_like(rewards) * -1, rewards)

    # reset if drawer is open or max length reached
    successes = torch.where(cabinet_dof_pos[:, 3] > 0.39, torch.ones_like(successes), successes)
    goal_reach = torch.where(cabinet_dof_pos[:, 3] > 0.39, torch.ones_like(reset_goal_buf),
    ↪   torch.zeros_like(reset_goal_buf))
    reset_buf = torch.where(progress_buf >= max_episode_length - 1, torch.ones_like(reset_buf),
    ↪   reset_buf)

    consecutive_successes = torch.where(reset_buf > 0, successes * reset_buf, consecutive_successes)
    heuristic_reward, task_reward = rewards, goal_reach
    return heuristic_reward, reset_buf, task_reward, consecutive_successes
```

```python
def compute_franka_reward(
    reset_buf, progress_buf, reset_goal_buf, actions, states, reward_settings, max_episode_length
):
    # type: (Tensor, Tensor, Tensor, Tensor, Dict[str, Tensor], Dict[str, float], float) -> Tuple[Tensor,
    ↪   Tensor, Tensor]

    # Compute per-env physical parameters
    target_height = states["cubeB_size"] + states["cubeA_size"] / 2.0
    cubeA_size = states["cubeA_size"]
    cubeB_size = states["cubeB_size"]

    # distance from hand to the cubeA
    d = torch.norm(states["cubeA_pos_relative"], dim=-1)
    d_lf = torch.norm(states["cubeA_pos"] - states["eef_lf_pos"], dim=-1)
    d_rf = torch.norm(states["cubeA_pos"] - states["eef_rf_pos"], dim=-1)
    dist_reward = 1 - torch.tanh(10.0 * (d + d_lf + d_rf) / 3)

    # reward for lifting cubeA
    cubeA_height = states["cubeA_pos"][:, 2] - reward_settings["table_height"]
    cubeA_lifted = (cubeA_height - cubeA_size) > 0.04
    lift_reward = cubeA_lifted

    # how closely aligned cubeA is to cubeB (only provided if cubeA is lifted)
    offset = torch.zeros_like(states["cubeA_to_cubeB_pos"])
    offset[:, 2] = (cubeA_size + cubeB_size) / 2
    d_ab = torch.norm(states["cubeA_to_cubeB_pos"] + offset, dim=-1)
    align_reward = (1 - torch.tanh(10.0 * d_ab)) * cubeA_lifted

    # Dist reward is maximum of dist and align reward
    dist_reward = torch.max(dist_reward, align_reward)

    # final reward for stacking successfully (only if cubeA is close to target height and corresponding
    ↪   location, and gripper is not grasping)
    cubeA_align_cubeB = (torch.norm(states["cubeA_to_cubeB_pos"][:, :2], dim=-1) < 0.02)
    cubeA_on_cubeB = torch.abs(cubeA_height - target_height) < 0.02
    gripper_away_from_cubeA = (d > 0.04)
    stack_reward = cubeA_align_cubeB & cubeA_on_cubeB & gripper_away_from_cubeA

    # Compose rewards

    # We either provide the stack reward or the align + dist reward
    rewards = torch.where(
        stack_reward,
        reward_settings["r_stack_scale"] * stack_reward,
        reward_settings["r_dist_scale"] * dist_reward + reward_settings["r_lift_scale"] * lift_reward +
        ↪   reward_settings[
            "r_align_scale"] * align_reward,
    )

    # Compute resets
    reset_buf = torch.where((progress_buf >= max_episode_length - 1), torch.ones_like(reset_buf),
    ↪   reset_buf)
    goal_reach = torch.where(stack_reward > 0, torch.ones_like(reset_goal_buf),
    ↪   torch.zeros_like(reset_goal_buf))
    heuristic_reward, task_reward = rewards, goal_reach
    return heuristic_reward, task_reward, reset_buf
```

```python
def compute_hand_reward(
    rew_buf, reset_buf, reset_goal_buf, progress_buf, successes, consecutive_successes,
    max_episode_length: float, object_pos, object_rot, target_pos, target_rot,
    dist_reward_scale: float, rot_reward_scale: float, rot_eps: float,
    actions, action_penalty_scale: float,
    success_tolerance: float, reach_goal_bonus: float, fall_dist: float,
    fall_penalty: float, max_consecutive_successes: int, av_factor: float, ignore_z_rot: bool
):
    # Distance from the hand to the object
    goal_dist = torch.norm(object_pos - target_pos, p=2, dim=-1)

    if ignore_z_rot:
        success_tolerance = 2.0 * success_tolerance

    # Orientation alignment for the cube in hand and goal cube
    quat_diff = quat_mul(object_rot, quat_conjugate(target_rot))
    rot_dist = 2.0 * torch.asin(torch.clamp(torch.norm(quat_diff[:, 0:3], p=2, dim=-1), max=1.0))

    dist_rew = goal_dist * dist_reward_scale
    rot_rew = 1.0/(torch.abs(rot_dist) + rot_eps) * rot_reward_scale

    action_penalty = torch.sum(actions ** 2, dim=-1)

    # Total reward is: position distance + orientation alignment + action regularization + success bonus
    ↪  + fall penalty
    reward = dist_rew + action_penalty * action_penalty_scale

    # Find out which envs hit the goal and update successes count
    goal_reach = torch.where(torch.abs(rot_dist) <= success_tolerance, torch.ones_like(reset_goal_buf),
    ↪  reset_goal_buf)
    successes = successes + goal_reach

    # Success bonus: orientation is within `success_tolerance` of goal orientation
    reward = torch.where(goal_reach == 1, reward + reach_goal_bonus, reward)

    # Fall penalty: distance to the goal is larger than a threshold
    reward = torch.where(goal_dist >= fall_dist, reward + fall_penalty, reward)

    # Check env termination conditions, including maximum success number
    resets = torch.where(goal_dist >= fall_dist, torch.ones_like(reset_buf), reset_buf)
    if max_consecutive_successes > 0:
        # Reset progress buffer on goal envs if max_consecutive_successes > 0
        progress_buf = torch.where(torch.abs(rot_dist) <= success_tolerance,
        ↪  torch.zeros_like(progress_buf), progress_buf)
        resets = torch.where(successes >= max_consecutive_successes, torch.ones_like(resets), resets)
    resets = torch.where(progress_buf >= max_episode_length - 1, torch.ones_like(resets), resets)

    # Apply penalty for not reaching the goal
    if max_consecutive_successes > 0:
        reward = torch.where(progress_buf >= max_episode_length - 1, reward + 0.5 * fall_penalty, reward)

    num_resets = torch.sum(resets)
    finished_cons_successes = torch.sum(successes * resets.float())

    cons_successes = torch.where(num_resets > 0, av_factor*finished_cons_successes/num_resets + (1.0 -
    ↪  av_factor)*consecutive_successes, consecutive_successes)
    heuristic_reward, task_reward = reward, goal_reach
    return heuristic_reward, resets, task_reward, progress_buf, successes, cons_successes
```

## Isaac Gym - Manipulation Task: *AllegroHand*

```python
def compute_hand_reward(
    rew_buf, reset_buf, reset_goal_buf, progress_buf, successes, consecutive_successes,
    max_episode_length: float, object_pos, object_rot, target_pos, target_rot,
    dist_reward_scale: float, rot_reward_scale: float, rot_eps: float,
    actions, action_penalty_scale: float,
    success_tolerance: float, reach_goal_bonus: float, fall_dist: float,
    fall_penalty: float, max_consecutive_successes: int, av_factor: float, ignore_z_rot: bool
):
    # Distance from the hand to the object
    goal_dist = torch.norm(object_pos - target_pos, p=2, dim=-1)

    if ignore_z_rot:
        success_tolerance = 2.0 * success_tolerance

    # Orientation alignment for the cube in hand and goal cube
    quat_diff = quat_mul(object_rot, quat_conjugate(target_rot))
    rot_dist = 2.0 * torch.asin(torch.clamp(torch.norm(quat_diff[:, 0:3], p=2, dim=-1), max=1.0))

    dist_rew = goal_dist * dist_reward_scale
    rot_rew = 1.0/(torch.abs(rot_dist) + rot_eps) * rot_reward_scale

    action_penalty = torch.sum(actions ** 2, dim=-1)

    # Total reward is: position distance + orientation alignment + action regularization + success bonus
    # ↪  + fall penalty
    reward = dist_rew + rot_rew + action_penalty * action_penalty_scale

    # Find out which envs hit the goal and update successes count
    goal_reach = torch.where(torch.abs(rot_dist) <= success_tolerance, torch.ones_like(reset_goal_buf),
    ↪  reset_goal_buf)
    successes = successes + goal_reach

    # Success bonus: orientation is within `success_tolerance` of goal orientation
    reward = torch.where(goal_reach == 1, reward + reach_goal_bonus, reward)

    # Fall penalty: distance to the goal is larger than a threshold
    reward = torch.where(goal_dist >= fall_dist, reward + fall_penalty, reward)

    # Check env termination conditions, including maximum success number
    resets = torch.where(goal_dist >= fall_dist, torch.ones_like(reset_buf), reset_buf)
    if max_consecutive_successes > 0:
        # Reset progress buffer on goal envs if max_consecutive_successes > 0
        progress_buf = torch.where(torch.abs(rot_dist) <= success_tolerance,
        ↪  torch.zeros_like(progress_buf), progress_buf)
        resets = torch.where(successes >= max_consecutive_successes, torch.ones_like(resets), resets)

    timed_out = progress_buf >= max_episode_length - 1
    resets = torch.where(timed_out, torch.ones_like(resets), resets)

    # Apply penalty for not reaching the goal
    if max_consecutive_successes > 0:
        reward = torch.where(timed_out, reward + 0.5 * fall_penalty, reward)

    num_resets = torch.sum(resets)
    finished_cons_successes = torch.sum(successes * resets.float())

    cons_successes = torch.where(num_resets > 0, av_factor*finished_cons_successes/num_resets + (1.0 -
    ↪  av_factor)*consecutive_successes, consecutive_successes)
    heuristic_reward, task_reward = reward, goal_reach
    return heuristic_reward, resets, task_reward, progress_buf, successes, cons_successes
```

```python
def compute_hand_reward(
    rew_buf, reset_buf, reset_goal_buf, progress_buf, successes, consecutive_successes,
    max_episode_length: float, object_pos, object_rot, target_pos, target_rot,
    dist_reward_scale: float, rot_reward_scale: float, rot_eps: float,
    actions, action_penalty_scale: float,
    success_tolerance: float, reach_goal_bonus: float, fall_dist: float,
    fall_penalty: float, max_consecutive_successes: int, av_factor: float, ignore_z_rot: bool
):
    # Distance from the hand to the object
    goal_dist = torch.norm(target_pos - object_pos, p=2, dim=-1)
    if ignore_z_rot:
        success_tolerance = 2.0 * success_tolerance

    # Orientation alignment for the cube in hand and goal cube
    quat_diff = quat_mul(object_rot, quat_conjugate(target_rot))
    rot_dist = 2.0 * torch.asin(torch.clamp(torch.norm(quat_diff[:, 0:3], p=2, dim=-1), max=1.0))

    dist_rew = goal_dist

    # Total reward is: position distance + orientation alignment + action regularization + success bonus
    ↪  + fall penalty
    reward = torch.exp(-0.2*(dist_rew * dist_reward_scale + rot_dist))

    # Find out which envs hit the goal and update successes count
    goal_resets = torch.where(torch.abs(goal_dist) <= 0, torch.ones_like(reset_goal_buf), reset_goal_buf)
    successes = torch.where(successes == 0,
                    torch.where(goal_dist < 0.03, torch.ones_like(successes), successes), successes)

    # Success bonus: orientation is within `success_tolerance` of goal orientation
    reward = torch.where(goal_resets == 1, reward + reach_goal_bonus, reward)

    # Fall penalty: distance to the goal is larger than a threashold
    reward = torch.where(object_pos[:, 2] <= 0.2, reward + fall_penalty, reward)

    # Check env termination conditions, including maximum success number
    resets = torch.where(object_pos[:, 2] <= 0.2, torch.ones_like(reset_buf), reset_buf)
    if max_consecutive_successes > 0:
        # Reset progress buffer on goal envs if max_consecutive_successes > 0
        progress_buf = torch.where(torch.abs(rot_dist) <= success_tolerance,
        ↪  torch.zeros_like(progress_buf), progress_buf)
        resets = torch.where(successes >= max_consecutive_successes, torch.ones_like(resets), resets)
    resets = torch.where(progress_buf >= max_episode_length, torch.ones_like(resets), resets)

    # Apply penalty for not reaching the goal
    if max_consecutive_successes > 0:
        reward = torch.where(progress_buf >= max_episode_length, reward + 0.5 * fall_penalty, reward)

    cons_successes = torch.where(resets > 0, successes * resets, consecutive_successes).mean()

    goal_reach = torch.where(goal_dist <= 0.03, torch.ones_like(successes), torch.zeros_like(successes))
    heuristic_reward, task_reward = reward, goal_reach
    return heuristic_reward, task_reward, resets, goal_resets, progress_buf, successes, cons_successes
```

## Bi-DexHands: *ShadowHandCatchUnderarm*

```python
def compute_hand_reward(
    rew_buf, reset_buf, reset_goal_buf, progress_buf, successes, consecutive_successes,
    max_episode_length: float, object_pos, object_rot, target_pos, target_rot,
    dist_reward_scale: float, rot_reward_scale: float, rot_eps: float,
    actions, action_penalty_scale: float,
    success_tolerance: float, reach_goal_bonus: float, fall_dist: float,
    fall_penalty: float, max_consecutive_successes: int, av_factor: float, ignore_z_rot: bool
):
    # Distance from the hand to the object
    goal_dist = torch.norm(target_pos - object_pos, p=2, dim=-1)

    if ignore_z_rot:
        success_tolerance = 2.0 * success_tolerance

    # Orientation alignment for the cube in hand and goal cube
    quat_diff = quat_mul(object_rot, quat_conjugate(target_rot))
    rot_dist = 2.0 * torch.asin(torch.clamp(torch.norm(quat_diff[:, 0:3], p=2, dim=-1), max=1.0))

    dist_rew = goal_dist

    # Total reward is: position distance + orientation alignment + action regularization + success bonus
    #  + fall penalty
    reward = torch.exp(-0.2*(dist_rew * dist_reward_scale + rot_dist))

    # Find out which envs hit the goal and update successes count
    goal_resets = torch.where(torch.abs(goal_dist) <= 0, torch.ones_like(reset_goal_buf), reset_goal_buf)
    successes = torch.where(successes == 0,
                    torch.where(goal_dist < 0.03, torch.ones_like(successes), successes), successes)

    # Fall penalty: distance to the goal is larger than a threashold
    reward = torch.where(object_pos[:, 2] <= 0.1, reward + fall_penalty, reward)

    # Check env termination conditions, including maximum success number
    resets = torch.where(object_pos[:, 2] <= 0.1, torch.ones_like(reset_buf), reset_buf)
    if max_consecutive_successes > 0:
        # Reset progress buffer on goal envs if max_consecutive_successes > 0
        progress_buf = torch.where(torch.abs(rot_dist) <= success_tolerance,
        ↪  torch.zeros_like(progress_buf), progress_buf)
        resets = torch.where(successes >= max_consecutive_successes, torch.ones_like(resets), resets)
    resets = torch.where(progress_buf >= max_episode_length, torch.ones_like(resets), resets)

    # Apply penalty for not reaching the goal
    if max_consecutive_successes > 0:
        reward = torch.where(progress_buf >= max_episode_length, reward + 0.5 * fall_penalty, reward)

    cons_successes = torch.where(resets > 0, successes * resets, consecutive_successes)
    goal_reach = torch.where(goal_dist <= 0.03,
                            torch.ones_like(successes), torch.zeros_like(successes))
    heuristic_reward, task_reward = reward, goal_reach
    return heuristic_reward, task_reward, resets, goal_resets, progress_buf, successes, cons_successes
```

```python
def compute_hand_reward(
    rew_buf, reset_buf, reset_goal_buf, progress_buf, successes, consecutive_successes,
    max_episode_length: float, object_pos, object_rot, target_pos, target_rot, left_hand_base_pos,
    ↪  right_hand_base_pos,
    dist_reward_scale: float, rot_reward_scale: float, rot_eps: float,
    actions, action_penalty_scale: float,
    success_tolerance: float, reach_goal_bonus: float, fall_dist: float,
    fall_penalty: float, max_consecutive_successes: int, av_factor: float, ignore_z_rot: bool, device:
    ↪  str
):
    # Distance from the hand to the object
    goal_dist = torch.norm(target_pos - object_pos, p=2, dim=-1)
    if ignore_z_rot:
        success_tolerance = 2.0 * success_tolerance

    # Orientation alignment for the cube in hand and goal cube
    quat_diff = quat_mul(object_rot, quat_conjugate(target_rot))
    rot_dist = 2.0 * torch.asin(torch.clamp(torch.norm(quat_diff[:, 0:3], p=2, dim=-1), max=1.0))

    dist_rew = goal_dist

    # Total reward is: position distance + orientation alignment + action regularization + success bonus
    ↪  + fall penalty
    reward = torch.exp(-0.2*(dist_rew * dist_reward_scale + rot_dist))

    # Find out which envs hit the goal and update successes count
    goal_resets = torch.where(torch.abs(goal_dist) <= 0, torch.ones_like(reset_goal_buf), reset_goal_buf)
    successes = torch.where(successes == 0,
                    torch.where(goal_dist < 0.03, torch.ones_like(successes), successes), successes)

    # Check env termination conditions, including maximum success number
    right_hand_base_dist = torch.norm(right_hand_base_pos - torch.tensor([0.0, 0.0, 0.5],
    ↪  dtype=torch.float, device=device), p=2, dim=-1)
    left_hand_base_dist = torch.norm(left_hand_base_pos - torch.tensor([0.0, -0.8, 0.5],
    ↪  dtype=torch.float, device=device), p=2, dim=-1)

    resets = torch.where(right_hand_base_dist >= 0.1, torch.ones_like(reset_buf), reset_buf)
    resets = torch.where(left_hand_base_dist >= 0.1, torch.ones_like(resets), resets)
    resets = torch.where(object_pos[:, 2] <= 0.3, torch.ones_like(resets), resets)
    if max_consecutive_successes > 0:
        # Reset progress buffer on goal envs if max_consecutive_successes > 0
        progress_buf = torch.where(torch.abs(rot_dist) <= success_tolerance,
        ↪  torch.zeros_like(progress_buf), progress_buf)
        resets = torch.where(successes >= max_consecutive_successes, torch.ones_like(resets), resets)
    resets = torch.where(progress_buf >= max_episode_length, torch.ones_like(resets), resets)

    # Apply penalty for not reaching the goal
    if max_consecutive_successes > 0:
        reward = torch.where(progress_buf >= max_episode_length, reward + 0.5 * fall_penalty, reward)

    cons_successes = torch.where(resets > 0, successes * resets, consecutive_successes)
    goal_reach = torch.where(goal_dist <= 0.03,
                        torch.ones_like(successes), torch.zeros_like(successes))
    heuristic_reward, task_reward = reward, goal_reach
    return heuristic_reward, task_reward, resets, goal_resets, progress_buf, successes, cons_successes
```

# Bi-DexHands: *ShadowHandCatchAbreast*

```python
def compute_hand_reward(
    rew_buf, reset_buf, reset_goal_buf, progress_buf, successes, consecutive_successes,
    max_episode_length: float, object_pos, object_rot, target_pos, target_rot, left_hand_pos,
    ↪  right_hand_pos, left_hand_base_pos, right_hand_base_pos,
    dist_reward_scale: float, rot_reward_scale: float, rot_eps: float,
    actions, action_penalty_scale: float,
    success_tolerance: float, reach_goal_bonus: float, fall_dist: float,
    fall_penalty: float, max_consecutive_successes: int, av_factor: float, ignore_z_rot: bool, device:
    ↪  str
):
    # Distance from the hand to the object
    goal_dist = torch.norm(target_pos - object_pos, p=2, dim=-1)

    if ignore_z_rot:
        success_tolerance = 2.0 * success_tolerance

    # Orientation alignment for the cube in hand and goal cube
    quat_diff = quat_mul(object_rot, quat_conjugate(target_rot))
    rot_dist = 2.0 * torch.asin(torch.clamp(torch.norm(quat_diff[:, 0:3], p=2, dim=-1), max=1.0))

    dist_rew = goal_dist

    # Total reward is: position distance + orientation alignment + action regularization + success bonus
    ↪  + fall penalty
    reward = torch.exp(-0.2*(dist_rew * dist_reward_scale + rot_dist))

    # Find out which envs hit the goal and update successes count
    goal_resets = torch.where(torch.abs(goal_dist) <= 0, torch.ones_like(reset_goal_buf), reset_goal_buf)

    successes = torch.where(successes == 0,
                    torch.where(goal_dist < 0.03, torch.ones_like(successes), successes), successes)

    # Fall penalty: distance to the goal is larger than a threashold
    reward = torch.where(object_pos[:, 2] <= 0.2, reward + fall_penalty, reward)

    # Check env termination conditions, including maximum success number
    right_hand_base_dist = torch.norm(right_hand_base_pos - torch.tensor([-0.3, -0.55, 0.5],
    ↪  dtype=torch.float, device=device), p=2, dim=-1)
    left_hand_base_dist = torch.norm(left_hand_base_pos - torch.tensor([-0.3, -1.15, 0.5],
    ↪  dtype=torch.float, device=device), p=2, dim=-1)

    resets = torch.where(right_hand_base_dist >= 0.1, torch.ones_like(reset_buf), reset_buf)
    resets = torch.where(left_hand_base_dist >= 0.1, torch.ones_like(resets), resets)

    resets = torch.where(object_pos[:, 2] <= 0.2, torch.ones_like(resets), resets)
    if max_consecutive_successes > 0:
        # Reset progress buffer on goal envs if max_consecutive_successes > 0
        progress_buf = torch.where(torch.abs(rot_dist) <= success_tolerance,
        ↪  torch.zeros_like(progress_buf), progress_buf)
        resets = torch.where(successes >= max_consecutive_successes, torch.ones_like(resets), resets)
    resets = torch.where(progress_buf >= max_episode_length, torch.ones_like(resets), resets)

    # Apply penalty for not reaching the goal
    if max_consecutive_successes > 0:
        reward = torch.where(progress_buf >= max_episode_length, reward + 0.5 * fall_penalty, reward)

    cons_successes = torch.where(resets > 0, successes * resets, consecutive_successes)
    goal_reach = torch.where(goal_dist <= 0.03,
                        torch.ones_like(successes), torch.zeros_like(successes))
    heuristic_reward, task_reward = reward, goal_reach
    return heuristic_reward, task_reward, resets, goal_resets, progress_buf, successes, cons_successes
```

## Bi-DexHands: *ShadowHandTwoCatchUnderarm*

```python
def compute_hand_reward(
    rew_buf, reset_buf, reset_goal_buf, progress_buf, successes, consecutive_successes,
    max_episode_length: float, object_pos, object_rot, target_pos, target_rot, object_another_pos,
    ↪  object_another_rot, target_another_pos, target_another_rot,
    dist_reward_scale: float, rot_reward_scale: float, rot_eps: float,
    actions, action_penalty_scale: float,
    success_tolerance: float, reach_goal_bonus: float, fall_dist: float,
    fall_penalty: float, max_consecutive_successes: int, av_factor: float, ignore_z_rot: bool
):
    # Distance from the hand to the object
    goal_dist = torch.norm(target_pos - object_pos, p=2, dim=-1)
    if ignore_z_rot:
        success_tolerance = 2.0 * success_tolerance

    goal_another_dist = torch.norm(target_another_pos - object_another_pos, p=2, dim=-1)
    if ignore_z_rot:
        success_tolerance = 2.0 * success_tolerance

    # Orientation alignment for the cube in hand and goal cube
    quat_diff = quat_mul(object_rot, quat_conjugate(target_rot))
    rot_dist = 2.0 * torch.asin(torch.clamp(torch.norm(quat_diff[:, 0:3], p=2, dim=-1), max=1.0))

    quat_another_diff = quat_mul(object_another_rot, quat_conjugate(target_another_rot))
    rot_another_dist = 2.0 * torch.asin(torch.clamp(torch.norm(quat_another_diff[:, 0:3], p=2, dim=-1),
    ↪  max=1.0))

    dist_rew = goal_dist

    # Total reward is: position distance + orientation alignment + action regularization + success bonus
    ↪  + fall penalty
    reward = torch.exp(-0.2*(dist_rew * dist_reward_scale + rot_dist)) +
    ↪  torch.exp(-0.2*(goal_another_dist * dist_reward_scale + rot_another_dist))

    # Find out which envs hit the goal and update successes count
    goal_resets = torch.where(torch.abs(goal_dist) <= 0, torch.ones_like(reset_goal_buf), reset_goal_buf)
    successes = torch.where(successes == 0,
                    torch.where(goal_dist + goal_another_dist < 0.06, torch.ones_like(successes),
                    ↪  successes), successes)

    # Fall penalty: distance to the goal is larger than a threashold
    reward = torch.where(object_pos[:, 2] <= 0.2, reward + fall_penalty, reward)
    reward = torch.where(object_another_pos[:, 2] <= 0.2, reward + fall_penalty, reward)

    # Check env termination conditions, including maximum success number
    resets = torch.where(object_pos[:, 2] <= 0.2, torch.ones_like(reset_buf), reset_buf)
    resets = torch.where(object_another_pos[:, 2] <= 0.2, torch.ones_like(reset_buf), resets)

    if max_consecutive_successes > 0:
        # Reset progress buffer on goal envs if max_consecutive_successes > 0
        progress_buf = torch.where(torch.abs(rot_dist) <= success_tolerance,
        ↪  torch.zeros_like(progress_buf), progress_buf)
        resets = torch.where(successes >= max_consecutive_successes, torch.ones_like(resets), resets)
    resets = torch.where(progress_buf >= max_episode_length, torch.ones_like(resets), resets)

    # Apply penalty for not reaching the goal
    if max_consecutive_successes > 0:
        reward = torch.where(progress_buf >= max_episode_length, reward + 0.5 * fall_penalty, reward)

    cons_successes = torch.where(resets > 0, successes * resets, consecutive_successes).mean()
    goal_reach = torch.where(goal_dist + goal_another_dist <= 0.06,
                        torch.ones_like(successes), torch.zeros_like(successes))
    heuristic_reward, task_reward = reward, goal_reach
    return heuristic_reward, task_reward, resets, goal_resets, progress_buf, successes, cons_successes
```

# Bi-DexHands: *ShadowHandLiftUnderarm*

```python
def compute_hand_reward(
    rew_buf, reset_buf, reset_goal_buf, progress_buf, successes, consecutive_successes,
    max_episode_length: float, object_pos, object_rot, target_pos, target_rot, pot_left_handle_pos,
    ↪  pot_right_handle_pos,
    left_hand_pos, right_hand_pos,
    dist_reward_scale: float, rot_reward_scale: float, rot_eps: float,
    actions, action_penalty_scale: float,
    success_tolerance: float, reach_goal_bonus: float, fall_dist: float,
    fall_penalty: float, max_consecutive_successes: int, av_factor: float, ignore_z_rot: bool
):
    # Distance from the hand to the object
    goal_dist = torch.norm(target_pos - object_pos, p=2, dim=-1)

    # goal_dist = target_pos[:, 2] - object_pos[:, 2]
    right_hand_dist = torch.norm(pot_right_handle_pos - right_hand_pos, p=2, dim=-1)
    left_hand_dist = torch.norm(pot_left_handle_pos - left_hand_pos, p=2, dim=-1)

    right_hand_dist_rew = right_hand_dist
    left_hand_dist_rew = left_hand_dist

    # Total reward is: position distance + orientation alignment + action regularization + success bonus
    ↪  + fall penalty
    up_rew = torch.zeros_like(right_hand_dist_rew)
    up_rew = torch.where(right_hand_dist < 0.08,
                        torch.where(left_hand_dist < 0.08,
                                    3*(0.385 - goal_dist), up_rew), up_rew)

    reward = 0.2 - right_hand_dist_rew - left_hand_dist_rew + up_rew

    resets = torch.where(object_pos[:, 2] <= 0.3, torch.ones_like(reset_buf), reset_buf)
    resets = torch.where(right_hand_dist >= 0.2, torch.ones_like(resets), resets)
    resets = torch.where(left_hand_dist >= 0.2, torch.ones_like(resets), resets)

    # Find out which envs hit the goal and update successes count
    successes = torch.where(successes == 0,
                    torch.where(goal_dist < 0.05, torch.ones_like(successes), successes), successes)

    resets = torch.where(progress_buf >= max_episode_length, torch.ones_like(resets), resets)

    goal_resets = torch.zeros_like(resets)

    cons_successes = torch.where(resets > 0, successes * resets, consecutive_successes).mean()
    goal_reach = torch.where(goal_dist <= 0.05,
                            torch.ones_like(successes), torch.zeros_like(successes))
    heuristic_reward, task_reward = reward, goal_reach      return heuristic_reward, task_reward, resets,
    ↪  goal_resets, progress_buf, successes, cons_successes
```

```python
def compute_hand_reward(
    rew_buf, reset_buf, reset_goal_buf, progress_buf, successes, consecutive_successes,
    max_episode_length: float, object_pos, object_rot, target_pos, target_rot, door_left_handle_pos,
    ↪  door_right_handle_pos,
    left_hand_pos, right_hand_pos, right_hand_ff_pos, right_hand_mf_pos, right_hand_rf_pos,
    ↪  right_hand_lf_pos, right_hand_th_pos,
    left_hand_ff_pos, left_hand_mf_pos, left_hand_rf_pos, left_hand_lf_pos, left_hand_th_pos,
    dist_reward_scale: float, rot_reward_scale: float, rot_eps: float,
    actions, action_penalty_scale: float,
    success_tolerance: float, reach_goal_bonus: float, fall_dist: float,
    fall_penalty: float, max_consecutive_successes: int, av_factor: float, ignore_z_rot: bool
):
    # Distance from the hand to the object
    goal_dist = torch.norm(target_pos - object_pos, p=2, dim=-1)

    right_hand_finger_dist = (torch.norm(door_right_handle_pos - right_hand_ff_pos, p=2, dim=-1) +
    ↪  torch.norm(door_right_handle_pos - right_hand_mf_pos, p=2, dim=-1)
                            + torch.norm(door_right_handle_pos - right_hand_rf_pos, p=2, dim=-1) +
                            ↪  torch.norm(door_right_handle_pos - right_hand_lf_pos, p=2, dim=-1)
                            + torch.norm(door_right_handle_pos - right_hand_th_pos, p=2, dim=-1))
    left_hand_finger_dist = (torch.norm(door_left_handle_pos - left_hand_ff_pos, p=2, dim=-1) +
    ↪  torch.norm(door_left_handle_pos - left_hand_mf_pos, p=2, dim=-1)
                            + torch.norm(door_left_handle_pos - left_hand_rf_pos, p=2, dim=-1) +
                            ↪  torch.norm(door_left_handle_pos - left_hand_lf_pos, p=2, dim=-1)
                            + torch.norm(door_left_handle_pos - left_hand_th_pos, p=2, dim=-1))

    right_hand_dist_rew = right_hand_finger_dist
    left_hand_dist_rew = left_hand_finger_dist

    # Total reward is: position distance + orientation alignment + action regularization + success bonus
    ↪  + fall penalty
    up_rew = torch.zeros_like(right_hand_dist_rew)
    up_rew = torch.where(right_hand_finger_dist < 0.5,
                    torch.where(left_hand_finger_dist < 0.5,
                                torch.abs(door_right_handle_pos[:, 1] - door_left_handle_pos[:, 1]) *
                                ↪  2, up_rew), up_rew)

    reward = 2 - right_hand_dist_rew - left_hand_dist_rew + up_rew

    resets = torch.where(right_hand_finger_dist >= 1.5, torch.ones_like(reset_buf), reset_buf)
    resets = torch.where(left_hand_finger_dist >= 1.5, torch.ones_like(resets), resets)

    # Find out which envs hit the goal and update successes count
    successes = torch.where(successes == 0,
                    torch.where(torch.abs(door_right_handle_pos[:, 1] - door_left_handle_pos[:, 1]) >
                    ↪  0.5, torch.ones_like(successes), successes), successes)

    resets = torch.where(progress_buf >= max_episode_length, torch.ones_like(resets), resets)

    goal_resets = torch.zeros_like(resets)

    cons_successes = torch.where(resets > 0, successes * resets, consecutive_successes).mean()
    goal_reach = torch.where(torch.abs(door_right_handle_pos[:, 1] - door_left_handle_pos[:, 1]) >= 0.5,
                            torch.ones_like(successes), torch.zeros_like(successes))
    heuristic_reward, task_reward = reward, goal_reach       return heuristic_reward, task_reward, resets,
    ↪  goal_resets, progress_buf, successes, cons_successes
```

```python
def compute_hand_reward(
    rew_buf, reset_buf, reset_goal_buf, progress_buf, successes, consecutive_successes,
    max_episode_length: float, object_pos, object_rot, target_pos, target_rot, door_left_handle_pos,
    ↪  door_right_handle_pos,
    left_hand_pos, right_hand_pos, right_hand_ff_pos, right_hand_mf_pos, right_hand_rf_pos,
    ↪  right_hand_lf_pos, right_hand_th_pos,
    left_hand_ff_pos, left_hand_mf_pos, left_hand_rf_pos, left_hand_lf_pos, left_hand_th_pos,
    dist_reward_scale: float, rot_reward_scale: float, rot_eps: float,
    actions, action_penalty_scale: float,
    success_tolerance: float, reach_goal_bonus: float, fall_dist: float,
    fall_penalty: float, max_consecutive_successes: int, av_factor: float, ignore_z_rot: bool
):
    right_hand_finger_dist = (torch.norm(door_right_handle_pos - right_hand_ff_pos, p=2, dim=-1) +
    ↪  torch.norm(door_right_handle_pos - right_hand_mf_pos, p=2, dim=-1)
                            + torch.norm(door_right_handle_pos - right_hand_rf_pos, p=2, dim=-1) +
                            ↪  torch.norm(door_right_handle_pos - right_hand_lf_pos, p=2, dim=-1)
                            + torch.norm(door_right_handle_pos - right_hand_th_pos, p=2, dim=-1))
    left_hand_finger_dist = (torch.norm(door_left_handle_pos - left_hand_ff_pos, p=2, dim=-1) +
    ↪  torch.norm(door_left_handle_pos - left_hand_mf_pos, p=2, dim=-1)
                            + torch.norm(door_left_handle_pos - left_hand_rf_pos, p=2, dim=-1) +
                            ↪  torch.norm(door_left_handle_pos - left_hand_lf_pos, p=2, dim=-1)
                            + torch.norm(door_left_handle_pos - left_hand_th_pos, p=2, dim=-1))

    right_hand_dist_rew = right_hand_finger_dist
    left_hand_dist_rew = left_hand_finger_dist

    # Total reward is: position distance + orientation alignment + action regularization + success bonus
    ↪  + fall penalty
    up_rew = torch.zeros_like(right_hand_dist_rew)
    up_rew = torch.where(right_hand_finger_dist < 0.5,
                    torch.where(left_hand_finger_dist < 0.5,
                                torch.abs(door_right_handle_pos[:, 1] - door_left_handle_pos[:, 1]) *
                                ↪  2, up_rew), up_rew)
    reward = 2 - right_hand_dist_rew - left_hand_dist_rew + up_rew

    resets = torch.where(right_hand_finger_dist >= 1.5, torch.ones_like(reset_buf), reset_buf)
    resets = torch.where(left_hand_finger_dist >= 1.5, torch.ones_like(resets), resets)

    # Find out which envs hit the goal and update successes count
    successes = torch.where(successes == 0,
                    torch.where(torch.abs(door_right_handle_pos[:, 1] - door_left_handle_pos[:, 1]) >
                    ↪  0.5, torch.ones_like(successes), successes), successes)

    resets = torch.where(progress_buf >= max_episode_length, torch.ones_like(resets), resets)

    goal_resets = torch.zeros_like(resets)

    cons_successes = torch.where(resets > 0, successes * resets, consecutive_successes).mean()
    goal_reach = torch.where(torch.abs(door_right_handle_pos[:, 1] - door_left_handle_pos[:, 1]) >= 0.5,
                        torch.ones_like(successes), torch.zeros_like(successes))
    heuristic_reward, task_reward = reward, goal_reach     return heuristic_reward, task_reward, resets,
    ↪  goal_resets, progress_buf, successes, cons_successes
```

## Bi-DexHands: *ShadowHandDoorCloseInward*

```python
def compute_hand_reward(
    rew_buf, reset_buf, reset_goal_buf, progress_buf, successes, consecutive_successes,
    max_episode_length: float, object_pos, object_rot, target_pos, target_rot, door_left_handle_pos,
    ↪  door_right_handle_pos,
    left_hand_pos, right_hand_pos, right_hand_ff_pos, right_hand_mf_pos, right_hand_rf_pos,
    ↪  right_hand_lf_pos, right_hand_th_pos,
    left_hand_ff_pos, left_hand_mf_pos, left_hand_rf_pos, left_hand_lf_pos, left_hand_th_pos,
    dist_reward_scale: float, rot_reward_scale: float, rot_eps: float,
    actions, action_penalty_scale: float,
    success_tolerance: float, reach_goal_bonus: float, fall_dist: float,
    fall_penalty: float, max_consecutive_successes: int, av_factor: float, ignore_z_rot: bool
):
    right_hand_finger_dist = (torch.norm(door_right_handle_pos - right_hand_ff_pos, p=2, dim=-1) +
    ↪  torch.norm(door_right_handle_pos - right_hand_mf_pos, p=2, dim=-1)
                            + torch.norm(door_right_handle_pos - right_hand_rf_pos, p=2, dim=-1) +
                            ↪  torch.norm(door_right_handle_pos - right_hand_lf_pos, p=2, dim=-1)
                            + torch.norm(door_right_handle_pos - right_hand_th_pos, p=2, dim=-1))
    left_hand_finger_dist = (torch.norm(door_left_handle_pos - left_hand_ff_pos, p=2, dim=-1) +
    ↪  torch.norm(door_left_handle_pos - left_hand_mf_pos, p=2, dim=-1)
                            + torch.norm(door_left_handle_pos - left_hand_rf_pos, p=2, dim=-1) +
                            ↪  torch.norm(door_left_handle_pos - left_hand_lf_pos, p=2, dim=-1)
                            + torch.norm(door_left_handle_pos - left_hand_th_pos, p=2, dim=-1))

    right_hand_dist_rew = right_hand_finger_dist
    left_hand_dist_rew = left_hand_finger_dist

    # Total reward is: position distance + orientation alignment + action regularization + success bonus
    ↪  + fall penalty
    up_rew = torch.zeros_like(right_hand_dist_rew)
    up_rew = torch.where(right_hand_finger_dist < 0.5,
                    torch.where(left_hand_finger_dist < 0.5,
                                1 - torch.abs(door_right_handle_pos[:, 1] - door_left_handle_pos[:,
                                ↪  1]) * 2, up_rew), up_rew)

    reward = 2 - right_hand_dist_rew - left_hand_dist_rew + up_rew

    resets = torch.where(right_hand_finger_dist >= 1.5, torch.ones_like(reset_buf), reset_buf)
    resets = torch.where(left_hand_finger_dist >= 1.5, torch.ones_like(resets), resets)

    # Find out which envs hit the goal and update successes count
    successes = torch.where(successes == 0,
                    torch.where(torch.abs(door_right_handle_pos[:, 1] - door_left_handle_pos[:, 1]) <
                    ↪  0.5, torch.ones_like(successes), successes), successes)

    resets = torch.where(progress_buf >= max_episode_length, torch.ones_like(resets), resets)

    goal_resets = torch.zeros_like(resets)

    cons_successes = torch.where(resets > 0, successes * resets, consecutive_successes)
    goal_reach = torch.where(torch.abs(door_right_handle_pos[:, 1] - door_left_handle_pos[:, 1]) <= 0.5,
                            torch.ones_like(successes), torch.zeros_like(successes))
    heuristic_reward, task_reward = reward, goal_reach       return heuristic_reward, task_reward, resets,
    ↪  goal_resets, progress_buf, successes, cons_successes
```

## Bi-DexHands: *ShadowHandDoorCloseOutward*

```python
def compute_hand_reward(
    rew_buf, reset_buf, reset_goal_buf, progress_buf, successes, consecutive_successes,
    max_episode_length: float, object_pos, object_rot, target_pos, target_rot, door_left_handle_pos,
    ↪  door_right_handle_pos,
    left_hand_pos, right_hand_pos, right_hand_ff_pos, right_hand_mf_pos, right_hand_rf_pos,
    ↪  right_hand_lf_pos, right_hand_th_pos,
    left_hand_ff_pos, left_hand_mf_pos, left_hand_rf_pos, left_hand_lf_pos, left_hand_th_pos,
    dist_reward_scale: float, rot_reward_scale: float, rot_eps: float,
    actions, action_penalty_scale: float,
    success_tolerance: float, reach_goal_bonus: float, fall_dist: float,
    fall_penalty: float, max_consecutive_successes: int, av_factor: float, ignore_z_rot: bool
):
    right_hand_finger_dist = (torch.norm(door_right_handle_pos - right_hand_ff_pos, p=2, dim=-1) +
    ↪  torch.norm(door_right_handle_pos - right_hand_mf_pos, p=2, dim=-1)
                            + torch.norm(door_right_handle_pos - right_hand_rf_pos, p=2, dim=-1) +
                            ↪  torch.norm(door_right_handle_pos - right_hand_lf_pos, p=2, dim=-1)
                            + torch.norm(door_right_handle_pos - right_hand_th_pos, p=2, dim=-1))
    left_hand_finger_dist = (torch.norm(door_left_handle_pos - left_hand_ff_pos, p=2, dim=-1) +
    ↪  torch.norm(door_left_handle_pos - left_hand_mf_pos, p=2, dim=-1)
                            + torch.norm(door_left_handle_pos - left_hand_rf_pos, p=2, dim=-1) +
                            ↪  torch.norm(door_left_handle_pos - left_hand_lf_pos, p=2, dim=-1)
                            + torch.norm(door_left_handle_pos - left_hand_th_pos, p=2, dim=-1))

    right_hand_dist_rew = right_hand_finger_dist
    left_hand_dist_rew = left_hand_finger_dist

    # Total reward is: position distance + orientation alignment + action regularization + success bonus
    ↪  + fall penalty
    up_rew = torch.zeros_like(right_hand_dist_rew)
    up_rew = torch.where(right_hand_finger_dist < 0.5,
                    torch.where(left_hand_finger_dist < 0.5,
                                1 - torch.abs(door_right_handle_pos[:, 1] - door_left_handle_pos[:,
                                ↪  1]) * 2, up_rew), up_rew)

    reward = 6 - right_hand_dist_rew - left_hand_dist_rew + up_rew

    resets = torch.where(right_hand_finger_dist >= 3, torch.ones_like(reset_buf), reset_buf)
    resets = torch.where(left_hand_finger_dist >= 3, torch.ones_like(resets), resets)

    # Find out which envs hit the goal and update successes count
    successes = torch.where(successes == 0,
                    torch.where(torch.abs(door_right_handle_pos[:, 1] - door_left_handle_pos[:, 1]) <
                    ↪  0.5, torch.ones_like(successes), successes), successes)

    resets = torch.where(progress_buf >= max_episode_length, torch.ones_like(resets), resets)

    goal_resets = torch.zeros_like(resets)

    cons_successes = torch.where(resets > 0, successes * resets, consecutive_successes).mean()
    goal_reach = torch.where(torch.abs(door_right_handle_pos[:, 1] - door_left_handle_pos[:, 1]) <= 0.5,
                torch.ones_like(successes), torch.zeros_like(successes))
    heuristic_reward, task_reward = reward, goal_reach       return heuristic_reward, task_reward, resets,
    ↪  goal_resets, progress_buf, successes, cons_successes
```

# Bi-DexHands: *ShadowHandSpin*

```python
def compute_hand_reward(
    rew_buf, reset_buf, reset_goal_buf, progress_buf, successes, consecutive_successes,
    max_episode_length: float, object_pos, object_rot, target_pos, target_rot,
    dist_reward_scale: float, rot_reward_scale: float, rot_eps: float,
    actions, action_penalty_scale: float,
    success_tolerance: float, reach_goal_bonus: float, fall_dist: float,
    fall_penalty: float, max_consecutive_successes: int, av_factor: float, ignore_z_rot: bool
):
    # Distance from the hand to the object
    goal_dist = torch.norm(object_pos - target_pos, p=2, dim=-1)

    if ignore_z_rot:
        success_tolerance = 2.0 * success_tolerance

    # Orientation alignment
    # Modified so pen is symmetrical; since we only rotate around the z axis,
    quat_diff_1 = quat_mul(object_rot, quat_conjugate(target_rot))
    rot_dist_1 = 2.0 * torch.asin(torch.clamp(torch.norm(quat_diff_1[:, 0:3], p=2, dim=-1), max=1.0))
    quat_diff_2 = quat_mul(object_rot, quat_conjugate(flip_orientation(target_rot)))
    rot_dist_2 = 2.0 * torch.asin(torch.clamp(torch.norm(quat_diff_2[:, 0:3], p=2, dim=-1), max=1.0))
    rot_dist = torch.min(rot_dist_1, rot_dist_2)

    dist_rew = goal_dist * dist_reward_scale
    rot_rew = 1.0/(torch.abs(rot_dist) + rot_eps) * rot_reward_scale

    action_penalty = torch.sum(actions ** 2, dim=-1)

    # Total reward is: position distance + orientation alignment + action regularization + success bonus
    ↪  + fall penalty
    reward = dist_rew + rot_rew + action_penalty * action_penalty_scale

    # Find out which envs hit the goal and update successes count
    goal_resets = torch.where(torch.abs(rot_dist) <= success_tolerance, torch.ones_like(reset_goal_buf),
    ↪  reset_goal_buf)
    successes = successes + goal_resets

    # Success bonus: orientation is within `success_tolerance` of goal orientation
    reward = torch.where(goal_resets == 1, reward + reach_goal_bonus, reward)

    # Fall penalty: distance to the goal is larger than a threshold
    reward = torch.where(goal_dist >= fall_dist, reward + fall_penalty, reward)

    # Check env termination conditions, including maximum success number
    resets = torch.where(goal_dist >= fall_dist, torch.ones_like(reset_buf), reset_buf)
    if max_consecutive_successes > 0:
        # Reset progress buffer on goal envs if max_consecutive_successes > 0
        progress_buf = torch.where(torch.abs(rot_dist) <= success_tolerance,
        ↪  torch.zeros_like(progress_buf), progress_buf)
        resets = torch.where(successes >= max_consecutive_successes, torch.ones_like(resets), resets)
    resets = torch.where(progress_buf >= max_episode_length - 1, torch.ones_like(resets), resets)

    # Apply penalty for not reaching the goal
    if max_consecutive_successes > 0:
        reward = torch.where(progress_buf >= max_episode_length - 1, reward + 0.5 * fall_penalty, reward)

    num_resets = torch.sum(resets)
    finished_cons_successes = torch.sum(successes * resets.float())

    cons_successes = torch.where(num_resets > 0, av_factor*finished_cons_successes/num_resets + (1.0 -
    ↪  av_factor)*consecutive_successes, consecutive_successes)
    goal_reach = torch.where(torch.abs(rot_dist) <= success_tolerance,
                             torch.ones_like(reset_goal_buf), torch.zeros_like(reset_goal_buf))
    heuristic_reward, task_reward = reward, goal_reach       return heuristic_reward, task_reward, resets,
    ↪  goal_resets, progress_buf, successes, cons_successes
```

# Bi-DexHands: *ShadowHandUpsideDown*

```python
def compute_hand_reward(
    rew_buf, reset_buf, reset_goal_buf, progress_buf, successes, consecutive_successes,
    max_episode_length: float, object_pos, object_rot, target_pos, target_rot,
    dist_reward_scale: float, rot_reward_scale: float, rot_eps: float,
    actions, action_penalty_scale: float,
    success_tolerance: float, reach_goal_bonus: float, fall_dist: float,
    fall_penalty: float, max_consecutive_successes: int, av_factor: float, ignore_z_rot: bool
):
    # Distance from the hand to the object
    goal_dist = torch.norm(object_pos - target_pos, p=2, dim=-1)

    if ignore_z_rot:
        success_tolerance = 2.0 * success_tolerance

    # Orientation alignment for the cube in hand and goal cube
    quat_diff = quat_mul(object_rot, quat_conjugate(target_rot))
    rot_dist = 2.0 * torch.asin(torch.clamp(torch.norm(quat_diff[:, 0:3], p=2, dim=-1), max=1.0))

    dist_rew = goal_dist * dist_reward_scale
    rot_rew = 1.0/(torch.abs(rot_dist) + rot_eps) * rot_reward_scale

    action_penalty = torch.sum(actions ** 2, dim=-1)

    # Total reward is: position distance + orientation alignment + action regularization + success bonus
    ↪  + fall penalty
    reward = dist_rew + rot_rew + action_penalty * action_penalty_scale

    # Find out which envs hit the goal and update successes count
    goal_resets = torch.where(torch.abs(rot_dist) <= success_tolerance, torch.ones_like(reset_goal_buf),
    ↪  reset_goal_buf)
    successes = successes + goal_resets

    # Success bonus: orientation is within `success_tolerance` of goal orientation
    reward = torch.where(goal_resets == 1, reward + reach_goal_bonus, reward)

    # Fall penalty: distance to the goal is larger than a threshold
    reward = torch.where(goal_dist >= fall_dist, reward + fall_penalty, reward)

    # Check env termination conditions, including maximum success number
    resets = torch.where(goal_dist >= fall_dist, torch.ones_like(reset_buf), reset_buf)
    if max_consecutive_successes > 0:
        # Reset progress buffer on goal envs if max_consecutive_successes > 0
        progress_buf = torch.where(torch.abs(rot_dist) <= success_tolerance,
        ↪  torch.zeros_like(progress_buf), progress_buf)
        resets = torch.where(successes >= max_consecutive_successes, torch.ones_like(resets), resets)
    resets = torch.where(progress_buf >= max_episode_length - 1, torch.ones_like(resets), resets)

    # Apply penalty for not reaching the goal
    if max_consecutive_successes > 0:
        reward = torch.where(progress_buf >= max_episode_length - 1, reward + 0.5 * fall_penalty, reward)

    num_resets = torch.sum(resets)
    finished_cons_successes = torch.sum(successes * resets.float())

    cons_successes = torch.where(num_resets > 0, av_factor*finished_cons_successes/num_resets + (1.0 -
    ↪  av_factor)*consecutive_successes, consecutive_successes)
    goal_reach = torch.where(torch.abs(rot_dist) <= success_tolerance, torch.ones_like(reset_goal_buf),
    ↪  torch.zeros_like(reset_goal_buf))
    heuristic_reward, task_reward = reward, goal_reach       return heuristic_reward, task_reward, resets,
    ↪  goal_resets, progress_buf, successes, cons_successes
```

# Bi-DexHands: *ShadowHandBlockStack*

```python
def compute_hand_reward(
    rew_buf, reset_buf, reset_goal_buf, progress_buf, successes, consecutive_successes,
    max_episode_length: float, object_pos, object_rot, target_pos, target_rot, block_right_handle_pos,
    ↪  block_left_handle_pos,
    left_hand_pos, right_hand_pos, right_hand_ff_pos, right_hand_mf_pos, right_hand_rf_pos,
    ↪  right_hand_lf_pos, right_hand_th_pos,
    left_hand_ff_pos, left_hand_mf_pos, left_hand_rf_pos, left_hand_lf_pos, left_hand_th_pos,
    dist_reward_scale: float, rot_reward_scale: float, rot_eps: float,
    actions, action_penalty_scale: float,
    success_tolerance: float, reach_goal_bonus: float, fall_dist: float,
    fall_penalty: float, max_consecutive_successes: int, av_factor: float, ignore_z_rot: bool
):
    # Distance from the hand to the object
    stack_pos1 = target_pos.clone()
    stack_pos2 = target_pos.clone()

    stack_pos1[:, 1] -= 0.1
    stack_pos2[:, 1] -= 0.1
    stack_pos1[:, 2] += 0.05

    goal_dist1 = torch.norm(stack_pos1 - block_left_handle_pos, p=2, dim=-1)
    goal_dist2 = torch.norm(stack_pos2 - block_right_handle_pos, p=2, dim=-1)

    right_hand_finger_dist = (torch.norm(block_right_handle_pos - right_hand_ff_pos, p=2, dim=-1) +
    ↪  torch.norm(block_right_handle_pos - right_hand_mf_pos, p=2, dim=-1)
                            + torch.norm(block_right_handle_pos - right_hand_rf_pos, p=2, dim=-1) +
                            ↪  torch.norm(block_right_handle_pos - right_hand_lf_pos, p=2, dim=-1)
                            + torch.norm(block_right_handle_pos - right_hand_th_pos, p=2, dim=-1))
    left_hand_finger_dist = (torch.norm(block_left_handle_pos - left_hand_ff_pos, p=2, dim=-1) +
    ↪  torch.norm(block_left_handle_pos - left_hand_mf_pos, p=2, dim=-1)
                            + torch.norm(block_left_handle_pos - left_hand_rf_pos, p=2, dim=-1) +
                            ↪  torch.norm(block_left_handle_pos - left_hand_lf_pos, p=2, dim=-1)
                            + torch.norm(block_left_handle_pos - left_hand_th_pos, p=2, dim=-1))

    right_hand_dist_rew = right_hand_finger_dist
    left_hand_dist_rew = left_hand_finger_dist

    # Total reward is: position distance + orientation alignment + action regularization + success bonus
    ↪  + fall penalty
    up_rew = torch.zeros_like(right_hand_dist_rew)
    up_rew = torch.where(right_hand_finger_dist < 0.5,
                    torch.where(left_hand_finger_dist < 0.5,
                        (0.24 - goal_dist1 - goal_dist2) * 2, up_rew), up_rew)

    stack_rew = torch.zeros_like(right_hand_dist_rew)
    stack_rew = torch.where(goal_dist2 < 0.07,
                    torch.where(goal_dist1 < 0.07,
                        (0.05-torch.abs(stack_pos1[:, 2] - block_left_handle_pos[:, 2])) * 50
                        ↪  ,stack_rew),stack_rew)

    reward = 1.5 - right_hand_dist_rew - left_hand_dist_rew + up_rew + stack_rew

    resets = torch.where(right_hand_dist_rew <= 0, torch.ones_like(reset_buf), reset_buf)
    resets = torch.where(right_hand_finger_dist >= 0.75, torch.ones_like(resets), resets)
    resets = torch.where(left_hand_finger_dist >= 0.75, torch.ones_like(resets), resets)

    # Find out which envs hit the goal and update successes count
    successes = torch.where(successes == 0,
                    torch.where(stack_rew > 1, torch.ones_like(successes), successes), successes)

    resets = torch.where(progress_buf >= max_episode_length, torch.ones_like(resets), resets)

    goal_resets = torch.zeros_like(resets)

    cons_successes = torch.where(resets > 0, successes * resets, consecutive_successes)
    goal_reach = torch.where(stack_rew >= 1,
                            torch.ones_like(successes), torch.zeros_like(successes))
    heuristic_reward, task_reward = reward, goal_reach      return heuristic_reward, task_reward, resets,
    ↪  goal_resets, progress_buf, successes, cons_successes
```

# Bi-DexHands: *ShadowHandBottleCap*

```python
def compute_hand_reward(
    rew_buf, reset_buf, reset_goal_buf, progress_buf, successes, consecutive_successes,
    max_episode_length: float, object_pos, object_rot, target_pos, target_rot, bottle_cap_pos,
    ↪  bottle_pos, bottle_cap_up,
    left_hand_pos, right_hand_pos, right_hand_ff_pos, right_hand_mf_pos, right_hand_rf_pos,
    ↪  right_hand_lf_pos, right_hand_th_pos,
    dist_reward_scale: float, rot_reward_scale: float, rot_eps: float,
    actions, action_penalty_scale: float,
    success_tolerance: float, reach_goal_bonus: float, fall_dist: float,
    fall_penalty: float, max_consecutive_successes: int, av_factor: float, ignore_z_rot: bool
):
    right_hand_dist = torch.norm(bottle_cap_pos - right_hand_pos, p=2, dim=-1)
    left_hand_dist = torch.norm(bottle_pos - left_hand_pos, p=2, dim=-1)

    right_hand_finger_dist = (torch.norm(bottle_cap_pos - right_hand_ff_pos, p=2, dim=-1) +
    ↪  torch.norm(bottle_cap_pos - right_hand_mf_pos, p=2, dim=-1)
                            + torch.norm(bottle_cap_pos - right_hand_rf_pos, p=2, dim=-1) +
                            ↪  torch.norm(bottle_cap_pos - right_hand_lf_pos, p=2, dim=-1)
                            + torch.norm(bottle_cap_pos - right_hand_th_pos, p=2, dim=-1))

    right_hand_dist_rew = right_hand_finger_dist
    left_hand_dist_rew = left_hand_dist

    # Total reward is: position distance + orientation alignment + action regularization + success bonus
    ↪  + fall penalty
    up_rew = torch.zeros_like(right_hand_dist_rew)

    up_rew =  torch.where(right_hand_finger_dist <= 0.3, torch.norm(bottle_cap_up - bottle_pos, p=2,
    ↪  dim=-1) * 30, up_rew)

    reward = 2.0 - right_hand_dist_rew - left_hand_dist_rew + up_rew

    resets = torch.where(bottle_cap_pos[:, 2] <= 0.5, torch.ones_like(reset_buf), reset_buf)
    resets = torch.where(right_hand_dist >= 0.5, torch.ones_like(resets), resets)
    resets = torch.where(left_hand_dist >= 0.2, torch.ones_like(resets), resets)

    # Find out which envs hit the goal and update successes count
    successes = torch.where(successes == 0,
                    torch.where(torch.norm(bottle_cap_up - bottle_pos, p=2, dim=-1) > 0.03,
                    ↪  torch.ones_like(successes), successes), successes)

    resets = torch.where(progress_buf >= max_episode_length, torch.ones_like(resets), resets)

    goal_resets = torch.zeros_like(resets)

    cons_successes = torch.where(resets > 0, successes * resets, consecutive_successes)
    goal_reach = torch.where(torch.norm(bottle_cap_up - bottle_pos, p=2, dim=-1) >= 0.03,
                        torch.ones_like(successes), torch.zeros_like(successes))
    heuristic_reward, task_reward = reward, goal_reach      return heuristic_reward, task_reward, resets,
    ↪  goal_resets, progress_buf, successes, cons_successes
```

## Bi-DexHands: *ShadowHandGraspAndPlace*

```python
def compute_hand_reward(
    rew_buf, reset_buf, reset_goal_buf, progress_buf, successes, consecutive_successes,
    max_episode_length: float, object_pos, object_rot, target_pos, target_rot, block_right_handle_pos,
    ↪  block_left_handle_pos,
    left_hand_pos, right_hand_pos, right_hand_ff_pos, right_hand_mf_pos, right_hand_rf_pos,
    ↪  right_hand_lf_pos, right_hand_th_pos,
    left_hand_ff_pos, left_hand_mf_pos, left_hand_rf_pos, left_hand_lf_pos, left_hand_th_pos,
    dist_reward_scale: float, rot_reward_scale: float, rot_eps: float,
    actions, action_penalty_scale: float,
    success_tolerance: float, reach_goal_bonus: float, fall_dist: float,
    fall_penalty: float, max_consecutive_successes: int, av_factor: float, ignore_z_rot: bool
):
    # Distance from the hand to the object
    right_hand_finger_dist = (torch.norm(block_right_handle_pos - right_hand_ff_pos, p=2, dim=-1) +
    ↪  torch.norm(block_right_handle_pos - right_hand_mf_pos, p=2, dim=-1)
                            + torch.norm(block_right_handle_pos - right_hand_rf_pos, p=2, dim=-1) +
                            ↪  torch.norm(block_right_handle_pos - right_hand_lf_pos, p=2, dim=-1)
                            + torch.norm(block_right_handle_pos - right_hand_th_pos, p=2, dim=-1))
    left_hand_finger_dist = (torch.norm(block_left_handle_pos - left_hand_ff_pos, p=2, dim=-1) +
    ↪  torch.norm(block_left_handle_pos - left_hand_mf_pos, p=2, dim=-1)
                            + torch.norm(block_left_handle_pos - left_hand_rf_pos, p=2, dim=-1) +
                            ↪  torch.norm(block_left_handle_pos - left_hand_lf_pos, p=2, dim=-1)
                            + torch.norm(block_left_handle_pos - left_hand_th_pos, p=2, dim=-1))

    right_hand_dist_rew = torch.exp(-10 * right_hand_finger_dist)
    left_hand_dist_rew = torch.exp(-10 * left_hand_finger_dist)

    up_rew = torch.zeros_like(right_hand_dist_rew)
    up_rew = torch.exp(-10 * torch.norm(block_right_handle_pos - block_left_handle_pos, p=2, dim=-1)) * 2

    reward = right_hand_dist_rew + left_hand_dist_rew + up_rew

    resets = torch.where(right_hand_dist_rew <= 0, torch.ones_like(reset_buf), reset_buf)
    resets = torch.where(right_hand_finger_dist >= 1.5, torch.ones_like(resets), resets)
    resets = torch.where(left_hand_finger_dist >= 1.5, torch.ones_like(resets), resets)

    # Find out which envs hit the goal and update successes count
    successes = torch.where(successes == 0,
                    torch.where(torch.norm(block_right_handle_pos - block_left_handle_pos, p=2, dim=-1) <
                    ↪  0.2, torch.ones_like(successes), successes), successes)

    resets = torch.where(progress_buf >= max_episode_length, torch.ones_like(resets), resets)

    goal_resets = torch.zeros_like(resets)

    cons_successes = torch.where(resets > 0, successes * resets, consecutive_successes).mean()
    goal_reach = torch.where(torch.norm(block_right_handle_pos - block_left_handle_pos, p=2, dim=-1) <=
    ↪  0.2,
                            torch.ones_like(successes), torch.zeros_like(successes))
    heuristic_reward, task_reward = reward, goal_reach      return heuristic_reward, task_reward, resets,
    ↪  goal_resets, progress_buf, successes, cons_successes
```

## Bi-DexHands: *ShadowHandKettle*

```python
def compute_hand_reward(
    rew_buf, reset_buf, reset_goal_buf, progress_buf, successes, consecutive_successes,
    max_episode_length: float, object_pos, object_rot, target_pos, target_rot, kettle_handle_pos,
    ↪  bucket_handle_pos, kettle_spout_pos,
    left_hand_pos, right_hand_pos, right_hand_ff_pos, right_hand_mf_pos, right_hand_rf_pos,
    ↪  right_hand_lf_pos, right_hand_th_pos,
    left_hand_ff_pos, left_hand_mf_pos, left_hand_rf_pos, left_hand_lf_pos, left_hand_th_pos,
    dist_reward_scale: float, rot_reward_scale: float, rot_eps: float,
    actions, action_penalty_scale: float,
    success_tolerance: float, reach_goal_bonus: float, fall_dist: float,
    fall_penalty: float, max_consecutive_successes: int, av_factor: float, ignore_z_rot: bool
):
    # Distance from the hand to the object
    right_hand_finger_dist = (torch.norm(kettle_handle_pos - right_hand_ff_pos, p=2, dim=-1) +
    ↪  torch.norm(kettle_handle_pos - right_hand_mf_pos, p=2, dim=-1)
                        + torch.norm(kettle_handle_pos - right_hand_rf_pos, p=2, dim=-1) +
                        ↪  torch.norm(kettle_handle_pos - right_hand_lf_pos, p=2, dim=-1)
                        + torch.norm(kettle_handle_pos - right_hand_th_pos, p=2, dim=-1))
    left_hand_finger_dist = (torch.norm(bucket_handle_pos - left_hand_ff_pos, p=2, dim=-1) +
    ↪  torch.norm(bucket_handle_pos - left_hand_mf_pos, p=2, dim=-1)
                        + torch.norm(bucket_handle_pos - left_hand_rf_pos, p=2, dim=-1) +
                        ↪  torch.norm(bucket_handle_pos - left_hand_lf_pos, p=2, dim=-1)
                        + torch.norm(bucket_handle_pos - left_hand_th_pos, p=2, dim=-1))

    right_hand_dist_rew = right_hand_finger_dist
    left_hand_dist_rew = left_hand_finger_dist

    # Total reward is: position distance + orientation alignment + action regularization + success bonus
    ↪  + fall penalty
    up_rew = torch.zeros_like(right_hand_dist_rew)
    up_rew = torch.where(right_hand_finger_dist < 0.7,
                    torch.where(left_hand_finger_dist < 0.7,
                                    0.5 - torch.norm(bucket_handle_pos - kettle_spout_pos, p=2, dim=-1) *
                                    ↪  2, up_rew), up_rew)

    reward = 1 + up_rew - right_hand_dist_rew - left_hand_dist_rew

    resets = torch.where(bucket_handle_pos[:, 2] <= 0.2, torch.ones_like(reset_buf), reset_buf)

    # Find out which envs hit the goal and update successes count
    successes = torch.where(successes == 0,
                    torch.where(torch.norm(bucket_handle_pos - kettle_spout_pos, p=2, dim=-1) < 0.05,
                    ↪  torch.ones_like(successes), successes), successes)

    resets = torch.where(progress_buf >= max_episode_length, torch.ones_like(resets), resets)

    goal_resets = torch.zeros_like(resets)

    cons_successes = torch.where(resets > 0, successes * resets, consecutive_successes).mean()
    goal_reach = torch.where(torch.norm(bucket_handle_pos - kettle_spout_pos, p=2, dim=-1) <= 0.05,
                        torch.ones_like(successes), torch.zeros_like(successes))
    heuristic_reward, task_reward = reward, goal_reach      return heuristic_reward, task_reward, resets,
    ↪  goal_resets, progress_buf, successes, cons_successes
```

# Bi-DexHands: *ShadowHandPen*

```python
def compute_hand_reward(
    rew_buf, reset_buf, reset_goal_buf, progress_buf, successes, consecutive_successes,
    max_episode_length: float, object_pos, object_rot, target_pos, target_rot, pen_right_handle_pos,
    ↪  pen_left_handle_pos,
    left_hand_pos, right_hand_pos, right_hand_ff_pos, right_hand_mf_pos, right_hand_rf_pos,
    ↪  right_hand_lf_pos, right_hand_th_pos,
    left_hand_ff_pos, left_hand_mf_pos, left_hand_rf_pos, left_hand_lf_pos, left_hand_th_pos,
    dist_reward_scale: float, rot_reward_scale: float, rot_eps: float,
    actions, action_penalty_scale: float,
    success_tolerance: float, reach_goal_bonus: float, fall_dist: float,
    fall_penalty: float, max_consecutive_successes: int, av_factor: float, ignore_z_rot: bool
):
    # Distance from the hand to the object
    right_hand_finger_dist = (torch.norm(pen_right_handle_pos - right_hand_ff_pos, p=2, dim=-1) +
    ↪  torch.norm(pen_right_handle_pos - right_hand_mf_pos, p=2, dim=-1)
                            + torch.norm(pen_right_handle_pos - right_hand_rf_pos, p=2, dim=-1) +
                            ↪  torch.norm(pen_right_handle_pos - right_hand_lf_pos, p=2, dim=-1)
                            + torch.norm(pen_right_handle_pos - right_hand_th_pos, p=2, dim=-1))
    left_hand_finger_dist = (torch.norm(pen_left_handle_pos - left_hand_ff_pos, p=2, dim=-1) +
    ↪  torch.norm(pen_left_handle_pos - left_hand_mf_pos, p=2, dim=-1)
                           + torch.norm(pen_left_handle_pos - left_hand_rf_pos, p=2, dim=-1) +
                           ↪  torch.norm(pen_left_handle_pos - left_hand_lf_pos, p=2, dim=-1)
                           + torch.norm(pen_left_handle_pos - left_hand_th_pos, p=2, dim=-1))

    right_hand_dist_rew = torch.exp(-10 * right_hand_finger_dist)
    left_hand_dist_rew = torch.exp(-10 * left_hand_finger_dist)

    # Total reward is: position distance + orientation alignment + action regularization + success bonus
    ↪  + fall penalty
    up_rew = torch.zeros_like(right_hand_dist_rew)
    up_rew = torch.where(right_hand_finger_dist < 0.75,
                    torch.where(left_hand_finger_dist < 0.75,
                        torch.norm(pen_right_handle_pos - pen_left_handle_pos, p=2, dim=-1) * 5 - 0.8,
                        ↪  up_rew), up_rew)

    reward = up_rew + right_hand_dist_rew + left_hand_dist_rew

    resets = torch.where(right_hand_dist_rew <= 0, torch.ones_like(reset_buf), reset_buf)
    resets = torch.where(right_hand_finger_dist >= 1.5, torch.ones_like(resets), resets)
    resets = torch.where(left_hand_finger_dist >= 1.5, torch.ones_like(resets), resets)

    # Find out which envs hit the goal and update successes count
    successes = torch.where(successes == 0,
                    torch.where(torch.norm(pen_right_handle_pos - pen_left_handle_pos, p=2, dim=-1) * 5 >
                    ↪  1.5, torch.ones_like(successes), successes), successes)

    resets = torch.where(progress_buf >= max_episode_length, torch.ones_like(resets), resets)

    goal_resets = torch.zeros_like(resets)

    cons_successes = torch.where(resets > 0, successes * resets, consecutive_successes)
    goal_reach = torch.where(torch.norm(pen_right_handle_pos - pen_left_handle_pos, p=2, dim=-1) * 5 >=
    ↪  1.5,
                        torch.ones_like(successes), torch.zeros_like(successes))
    heuristic_reward, task_reward = reward, goal_reach      return heuristic_reward, task_reward, resets,
    ↪  goal_resets, progress_buf, successes, cons_successes
```

# Bi-DexHands: *ShadowHandPushBlock*

```python
def compute_hand_reward(
    rew_buf, reset_buf, reset_goal_buf, progress_buf, successes, consecutive_successes,
    max_episode_length: float, object_pos, object_rot, left_target_pos, left_target_rot,
    ↪  right_target_pos, right_target_rot, block_right_handle_pos, block_left_handle_pos,
    left_hand_pos, right_hand_pos, right_hand_ff_pos, right_hand_mf_pos, right_hand_rf_pos,
    ↪  right_hand_lf_pos, right_hand_th_pos,
    left_hand_ff_pos, left_hand_mf_pos, left_hand_rf_pos, left_hand_lf_pos, left_hand_th_pos,
    dist_reward_scale: float, rot_reward_scale: float, rot_eps: float,
    actions, action_penalty_scale: float,
    success_tolerance: float, reach_goal_bonus: float, fall_dist: float,
    fall_penalty: float, max_consecutive_successes: int, av_factor: float, ignore_z_rot: bool
):
    # Distance from the hand to the object
    left_goal_dist = torch.norm(left_target_pos - block_left_handle_pos, p=2, dim=-1)
    right_goal_dist = torch.norm(right_target_pos - block_right_handle_pos, p=2, dim=-1)

    right_hand_finger_dist = (torch.norm(block_right_handle_pos - right_hand_ff_pos, p=2, dim=-1) +
    ↪  torch.norm(block_right_handle_pos - right_hand_mf_pos, p=2, dim=-1)
                            + torch.norm(block_right_handle_pos - right_hand_rf_pos, p=2, dim=-1) +
                            ↪  torch.norm(block_right_handle_pos - right_hand_lf_pos, p=2, dim=-1)
                            + torch.norm(block_right_handle_pos - right_hand_th_pos, p=2, dim=-1))
    left_hand_finger_dist = (torch.norm(block_left_handle_pos - left_hand_ff_pos, p=2, dim=-1) +
    ↪  torch.norm(block_left_handle_pos - left_hand_mf_pos, p=2, dim=-1)
                            + torch.norm(block_left_handle_pos - left_hand_rf_pos, p=2, dim=-1) +
                            ↪  torch.norm(block_left_handle_pos - left_hand_lf_pos, p=2, dim=-1)
                            + torch.norm(block_left_handle_pos - left_hand_th_pos, p=2, dim=-1))

    right_hand_dist_rew = 1.2-1*right_hand_finger_dist
    left_hand_dist_rew = 1.2-1*left_hand_finger_dist

    # Total reward is: position distance + orientation alignment + action regularization + success bonus
    ↪  + fall penalty
    up_rew = torch.zeros_like(right_hand_dist_rew)
    up_rew = 5 - 5*left_goal_dist - 5*right_goal_dist

    reward = right_hand_dist_rew + left_hand_dist_rew + up_rew

    resets = torch.where(right_hand_finger_dist >= 1.2, torch.ones_like(reset_buf), reset_buf)
    resets = torch.where(left_hand_finger_dist >= 1.2, torch.ones_like(resets), resets)

    # Find out which envs hit the goal and update successes count
    successes = torch.where(successes == 0,
                    torch.where(torch.abs(left_goal_dist) <= 0.1,
                        torch.where(torch.abs(right_goal_dist) <= 0.1, torch.ones_like(successes),
                        ↪  torch.ones_like(successes) * 0.5), successes), successes)

    resets = torch.where(progress_buf >= max_episode_length, torch.ones_like(resets), resets)

    goal_resets = torch.zeros_like(resets)

    cons_successes = torch.where(resets > 0, successes * resets, consecutive_successes).mean()
    goal_reach = 0.5 * (torch.where(torch.abs(left_goal_dist) <= 0.1,
                                torch.ones_like(successes), torch.zeros_like(successes)) \
                    + torch.where(torch.abs(right_goal_dist) <= 0.1,
                                torch.ones_like(successes), torch.zeros_like(successes)))
    heuristic_reward, task_reward = reward, goal_reach     return heuristic_reward, task_reward, resets,
    ↪  goal_resets, progress_buf, successes, cons_successes
```

# Bi-DexHands: *ShadowHandReOrientation*

```python
def compute_hand_reward(
    rew_buf, reset_buf, reset_goal_buf, progress_buf, successes, consecutive_successes,
    max_episode_length: float, object_pos, object_rot, target_pos, target_rot, object_another_pos,
    ↪  object_another_rot, target_another_pos, target_another_rot,
    dist_reward_scale: float, rot_reward_scale: float, rot_eps: float,
    actions, action_penalty_scale: float,
    success_tolerance: float, reach_goal_bonus: float, fall_dist: float,
    fall_penalty: float, max_consecutive_successes: int, av_factor: float, ignore_z_rot: bool
):
    # Distance from the hand to the object
    goal_dist = torch.norm(target_pos - object_pos, p=2, dim=-1)
    if ignore_z_rot:
        success_tolerance = 2.0 * success_tolerance

    goal_another_dist = torch.norm(target_another_pos - object_another_pos, p=2, dim=-1)
    if ignore_z_rot:
        success_tolerance = 2.0 * success_tolerance

    # Orientation alignment for the cube in hand and goal cube
    quat_diff = quat_mul(object_rot, quat_conjugate(target_rot))
    rot_dist = 2.0 * torch.asin(torch.clamp(torch.norm(quat_diff[:, 0:3], p=2, dim=-1), max=1.0))

    quat_another_diff = quat_mul(object_another_rot, quat_conjugate(target_another_rot))
    rot_another_dist = 2.0 * torch.asin(torch.clamp(torch.norm(quat_another_diff[:, 0:3], p=2, dim=-1),
    ↪   max=1.0))

    dist_rew = goal_dist * dist_reward_scale + goal_another_dist * dist_reward_scale
    rot_rew = 1.0/(torch.abs(rot_dist) + rot_eps) * rot_reward_scale + 1.0/(torch.abs(rot_another_dist) +
    ↪   rot_eps) * rot_reward_scale

    action_penalty = torch.sum(actions ** 2, dim=-1)

    # Total reward is: position distance + orientation alignment + action regularization + success bonus
    ↪   + fall penalty
    reward = dist_rew + rot_rew + action_penalty * action_penalty_scale

    # Find out which envs hit the goal and update successes count
    goal_resets = torch.where(torch.abs(rot_dist) < 0.1, torch.ones_like(reset_goal_buf), reset_goal_buf)
    goal_resets = torch.where(torch.abs(rot_another_dist) < 0.1, torch.ones_like(reset_goal_buf),
    ↪   reset_goal_buf)

    successes = successes + goal_resets

    # Success bonus: orientation is within `success_tolerance` of goal orientation
    reward = torch.where(goal_resets == 1, reward + reach_goal_bonus, reward)

    # Fall penalty: distance to the goal is larger than a threashold
    reward = torch.where(object_pos[:, 2] <= 0.2, reward + fall_penalty, reward)
    reward = torch.where(object_another_pos[:, 2] <= 0.2, reward + fall_penalty, reward)

    # Check env termination conditions, including maximum success number
    resets = torch.where(object_pos[:, 2] <= 0.2, torch.ones_like(reset_buf), reset_buf)
    resets = torch.where(object_another_pos[:, 2] <= 0.2, torch.ones_like(reset_buf), resets)

    if max_consecutive_successes > 0:
        # Reset progress buffer on goal envs if max_consecutive_successes > 0
        progress_buf = torch.where(torch.abs(rot_dist) <= success_tolerance,
        ↪   torch.zeros_like(progress_buf), progress_buf)
        resets = torch.where(successes >= max_consecutive_successes, torch.ones_like(resets), resets)
    resets = torch.where(progress_buf >= max_episode_length, torch.ones_like(resets), resets)

    # Apply penalty for not reaching the goal
    if max_consecutive_successes > 0:
        reward = torch.where(progress_buf >= max_episode_length, reward + 0.5 * fall_penalty, reward)

    num_resets = torch.sum(resets)
    finished_cons_successes = torch.sum(successes * resets.float())

    cons_successes = torch.where(num_resets > 0, av_factor*finished_cons_successes/num_resets + (1.0 -
    ↪   av_factor)*consecutive_successes, consecutive_successes)
    goal_reach = 0.5 * (torch.where(torch.abs(rot_dist) <= 0.1, torch.ones_like(reset_goal_buf),
    ↪   torch.zeros_like(reset_goal_buf)) \
            + torch.where(torch.abs(rot_another_dist) <= 0.1, torch.ones_like(reset_goal_buf),
            ↪   torch.zeros_like(reset_goal_buf)))
    heuristic_reward, task_reward = reward, goal_reach        return heuristic_reward, task_reward, resets,
    ↪   goal_resets, progress_buf, successes, cons_successes
```

# Bi-DexHands: *ShadowHandScissors*

```python
def compute_hand_reward(
    rew_buf, reset_buf, reset_goal_buf, progress_buf, successes, consecutive_successes,
    max_episode_length: float, object_pos, object_rot, target_pos, target_rot, scissors_right_handle_pos,
    ↪  scissors_left_handle_pos, object_dof_pos,
    left_hand_pos, right_hand_pos, right_hand_ff_pos, right_hand_mf_pos, right_hand_rf_pos,
    ↪  right_hand_lf_pos, right_hand_th_pos,
    left_hand_ff_pos, left_hand_mf_pos, left_hand_rf_pos, left_hand_lf_pos, left_hand_th_pos,
    dist_reward_scale: float, rot_reward_scale: float, rot_eps: float,
    actions, action_penalty_scale: float,
    success_tolerance: float, reach_goal_bonus: float, fall_dist: float,
    fall_penalty: float, max_consecutive_successes: int, av_factor: float, ignore_z_rot: bool
):
    # Distance from the hand to the object
    right_hand_finger_dist = (torch.norm(scissors_right_handle_pos - right_hand_ff_pos, p=2, dim=-1) +
    ↪  torch.norm(scissors_right_handle_pos - right_hand_mf_pos, p=2, dim=-1)
                             + torch.norm(scissors_right_handle_pos - right_hand_rf_pos, p=2, dim=-1) +
                             ↪  torch.norm(scissors_right_handle_pos - right_hand_lf_pos, p=2, dim=-1)
                             + torch.norm(scissors_right_handle_pos - right_hand_th_pos, p=2, dim=-1))
    left_hand_finger_dist = (torch.norm(scissors_left_handle_pos - left_hand_ff_pos, p=2, dim=-1) +
    ↪  torch.norm(scissors_left_handle_pos - left_hand_mf_pos, p=2, dim=-1)
                             + torch.norm(scissors_left_handle_pos - left_hand_rf_pos, p=2, dim=-1) +
                             ↪  torch.norm(scissors_left_handle_pos - left_hand_lf_pos, p=2, dim=-1)
                             + torch.norm(scissors_left_handle_pos - left_hand_th_pos, p=2, dim=-1))

    right_hand_dist_rew = right_hand_finger_dist
    left_hand_dist_rew = left_hand_finger_dist

    # Total reward is: position distance + orientation alignment + action regularization + success bonus
    ↪  + fall penalty
    up_rew = torch.zeros_like(right_hand_dist_rew)
    up_rew = torch.where(right_hand_finger_dist < 0.7,
                    torch.where(left_hand_finger_dist < 0.7,
                        (0.59 + object_dof_pos[:, 0]) * 5, up_rew), up_rew)

    reward = 2 + up_rew - right_hand_dist_rew - left_hand_dist_rew

    resets = torch.where(up_rew < -0.5, torch.ones_like(reset_buf), reset_buf)
    resets = torch.where(right_hand_finger_dist >= 1.75, torch.ones_like(resets), resets)
    resets = torch.where(left_hand_finger_dist >= 1.75, torch.ones_like(resets), resets)

    # Find out which envs hit the goal and update successes count
    resets = torch.where(progress_buf >= max_episode_length, torch.ones_like(resets), resets)

    successes = torch.where(successes == 0,
                    torch.where(object_dof_pos[:, 0] > -0.3, torch.ones_like(successes), successes),
                    ↪  successes)

    goal_resets = torch.zeros_like(resets)

    cons_successes = torch.where(resets > 0, successes * resets, consecutive_successes).mean()
    goal_reach = torch.where(object_dof_pos[:, 0] >= -0.3,
                        torch.ones_like(successes), torch.zeros_like(successes))
    heuristic_reward, task_reward = reward, goal_reach      return heuristic_reward, task_reward, resets,
    ↪  goal_resets, progress_buf, successes, cons_successes
```

## Bi-DexHands: *ShadowHandSwingCup*

```python
def compute_hand_reward(
    rew_buf, reset_buf, reset_goal_buf, progress_buf, successes, consecutive_successes,
    max_episode_length: float, object_pos, object_rot, target_pos, target_rot, cup_right_handle_pos,
    ↪  cup_left_handle_pos,
    left_hand_pos, right_hand_pos, right_hand_ff_pos, right_hand_mf_pos, right_hand_rf_pos,
    ↪  right_hand_lf_pos, right_hand_th_pos,
    left_hand_ff_pos, left_hand_mf_pos, left_hand_rf_pos, left_hand_lf_pos, left_hand_th_pos,
    dist_reward_scale: float, rot_reward_scale: float, rot_eps: float,
    actions, action_penalty_scale: float,
    success_tolerance: float, reach_goal_bonus: float, fall_dist: float,
    fall_penalty: float, max_consecutive_successes: int, av_factor: float, ignore_z_rot: bool
):
    # Distance from the hand to the object
    right_hand_finger_dist = (torch.norm(cup_right_handle_pos - right_hand_ff_pos, p=2, dim=-1) +
    ↪  torch.norm(cup_right_handle_pos - right_hand_mf_pos, p=2, dim=-1)
                            + torch.norm(cup_right_handle_pos - right_hand_rf_pos, p=2, dim=-1) +
                            ↪  torch.norm(cup_right_handle_pos - right_hand_lf_pos, p=2, dim=-1)
                            + torch.norm(cup_right_handle_pos - right_hand_th_pos, p=2, dim=-1))
    left_hand_finger_dist = (torch.norm(cup_left_handle_pos - left_hand_ff_pos, p=2, dim=-1) +
    ↪  torch.norm(cup_left_handle_pos - left_hand_mf_pos, p=2, dim=-1)
                            + torch.norm(cup_left_handle_pos - left_hand_rf_pos, p=2, dim=-1) +
                            ↪  torch.norm(cup_left_handle_pos - left_hand_lf_pos, p=2, dim=-1)
                            + torch.norm(cup_left_handle_pos - left_hand_th_pos, p=2, dim=-1))

    # Orientation alignment for the cube in hand and goal cube
    quat_diff = quat_mul(object_rot, quat_conjugate(target_rot))
    rot_dist = 2.0 * torch.asin(torch.clamp(torch.norm(quat_diff[:, 0:3], p=2, dim=-1), max=1.0))

    right_hand_dist_rew = right_hand_finger_dist
    left_hand_dist_rew = left_hand_finger_dist

    rot_rew = 1.0/(torch.abs(rot_dist) + rot_eps) * rot_reward_scale - 1

    # Total reward is: position distance + orientation alignment + action regularization + success bonus
    ↪  + fall penalty
    up_rew = torch.zeros_like(rot_rew)
    up_rew = torch.where(right_hand_finger_dist < 0.4,
                        torch.where(left_hand_finger_dist < 0.4,
                                    rot_rew, up_rew), up_rew)

    reward = - right_hand_dist_rew - left_hand_dist_rew + up_rew

    resets = torch.where(object_pos[:, 2] <= 0.3, torch.ones_like(reset_buf), reset_buf)

    # Find out which envs hit the goal and update successes count
    successes = torch.where(successes == 0,
                    torch.where(rot_dist < 0.785, torch.ones_like(successes), successes), successes)

    resets = torch.where(progress_buf >= max_episode_length, torch.ones_like(resets), resets)

    goal_resets = torch.zeros_like(resets)

    cons_successes = torch.where(resets > 0, successes * resets, consecutive_successes).mean()
    goal_reach = torch.where(rot_dist <= 0.785, torch.ones_like(successes), torch.zeros_like(successes))
    heuristic_reward, task_reward = reward, goal_reach      return heuristic_reward, task_reward, resets,
    ↪  goal_resets, progress_buf, successes, cons_successes
```

## Bi-DexHands: *ShadowHandSwitch*

```python
def compute_hand_reward(
    rew_buf, reset_buf, reset_goal_buf, progress_buf, successes, consecutive_successes,
    max_episode_length: float, object_pos, object_rot, target_pos, target_rot, switch_right_handle_pos,
    ↪  switch_left_handle_pos,
    left_hand_pos, right_hand_pos, right_hand_ff_pos, right_hand_mf_pos, right_hand_rf_pos,
    ↪  right_hand_lf_pos, right_hand_th_pos,
    left_hand_ff_pos, left_hand_mf_pos, left_hand_rf_pos, left_hand_lf_pos, left_hand_th_pos,
    dist_reward_scale: float, rot_reward_scale: float, rot_eps: float,
    actions, action_penalty_scale: float,
    success_tolerance: float, reach_goal_bonus: float, fall_dist: float,
    fall_penalty: float, max_consecutive_successes: int, av_factor: float, ignore_z_rot: bool
):
    # Distance from the hand to the object
    right_hand_finger_dist = (torch.norm(switch_right_handle_pos - right_hand_ff_pos, p=2, dim=-1) +
    ↪  torch.norm(switch_right_handle_pos - right_hand_mf_pos, p=2, dim=-1)
                             + torch.norm(switch_right_handle_pos - right_hand_rf_pos, p=2, dim=-1) +
                             ↪  torch.norm(switch_right_handle_pos - right_hand_lf_pos, p=2, dim=-1)
                             + torch.norm(switch_right_handle_pos - right_hand_th_pos, p=2, dim=-1))
    left_hand_finger_dist = (torch.norm(switch_left_handle_pos - left_hand_ff_pos, p=2, dim=-1) +
    ↪  torch.norm(switch_left_handle_pos - left_hand_mf_pos, p=2, dim=-1)
                             + torch.norm(switch_left_handle_pos - left_hand_rf_pos, p=2, dim=-1) +
                             ↪  torch.norm(switch_left_handle_pos - left_hand_lf_pos, p=2, dim=-1)
                             + torch.norm(switch_left_handle_pos - left_hand_th_pos, p=2, dim=-1))

    right_hand_dist_rew = right_hand_finger_dist
    left_hand_dist_rew = left_hand_finger_dist

    # Total reward is: position distance + orientation alignment + action regularization + success bonus
    ↪  + fall penalty
    up_rew = torch.zeros_like(right_hand_dist_rew)
    up_rew = (1.4-(switch_right_handle_pos[:, 2] + switch_left_handle_pos[:, 2])) * 50

    reward = 2 - right_hand_dist_rew - left_hand_dist_rew + up_rew

    resets = torch.where(right_hand_dist_rew <= 0, torch.ones_like(reset_buf), reset_buf)

    # Find out which envs hit the goal and update successes count
    successes = torch.where(successes == 0,
                    torch.where(1.4-(switch_right_handle_pos[:, 2] + switch_left_handle_pos[:, 2]) >
                    ↪  0.05, torch.ones_like(successes), successes), successes)

    resets = torch.where(progress_buf >= max_episode_length, torch.ones_like(resets), resets)

    goal_resets = torch.zeros_like(resets)

    cons_successes = torch.where(resets > 0, successes * resets, consecutive_successes).mean()
    goal_reach = torch.where(1.4 - (switch_right_handle_pos[:, 2] + switch_left_handle_pos[:, 2]) >=
    ↪  0.05,
                             torch.ones_like(successes), torch.zeros_like(successes))
    heuristic_reward, task_reward = reward, goal_reach    return heuristic_reward, task_reward, resets,
    ↪  goal_resets, progress_buf, successes, cons_successes
```

