# OpenReview forum: "Going Beyond Heuristics by Imposing Policy Improvement as a Constraint"
_NeurIPS.cc/2024/Conference — NeurIPS 2024 poster_

### Official Review · Reviewer_P4zd · 2024-07-05

**Soundness:** 4
**Presentation:** 4
**Contribution:** 3
**Rating:** 7
**Confidence:** 4

**Summary:**

In this paper, the authors address the problem of reward design in reinforcement learning. Specifically, it is common practice to add heuristics in the reward to help the training and a lot of manual engineering is needed to balance this heuristic and the main reward. They propose to modify the standard RL objective to a constrained optimization objective in order to incorporate heuristics in the reward function in a principled way.

Their formulation consists in optimizing both the task reward and the heuristic reward while constraining the policy to have a better performance in the task reward compared to a policy that is solely maximizing the heuristic reward. This constraint ensures that the policy is not only exploiting the heuristic but also solves the task.

The method is evaluated on a set of robotic control tasks where heuristics are already provided.

I acknowledge reading the authors rebuttal and updated my score.

**Strengths:**

They propose a simple formalization of the problem of reward design with adding heuristic to a main reward task. This does not address the general problem of reward alignment, but I think it is a very relevant problem in RL for domains with well defined but sparse rewards.

The derivation of the algorithm is presented clearly.

The experiments show that the proposed algorithm outperform the baseline in terms of average return. Except from the missing EIPO, the authors choose relevant baseline to compare against.

The second experiment of reward designed in the wild is interesting. The author attempt to recreate the process of reward design with human participants.

The statistical analysis of the experimental results is rigorous.

The ablation study shows the importance of using the policy trained on the heuristic reward as a reference.

**Weaknesses:**

The data collection part of the algorithm is not super clear. My understanding is that both policies are used and they collect the same amount of trajectories for each, why would the amount of data be the same and not more? The theoretical implications of this data sharing part should be clarified as well. It is not obvious to me that equation 7 and 8 actually lead to the correct policy improvement step.

No theoretical analysis of the algorithm is provided. I am not sure that detailed theorem and proofs would be needed but at least some insights or discussion about the convergence. There is a brief mention that it would not converge to the optimal policy in the limitations but no proof is given.

Why not include EIPO in the first experiment? Overall, there is no direct comparison with EIPO. The authors comment on certain aspects during the ablation study but it would have been even stronger to add a line for EIPO.

In the second experiment some results appear to be missing which makes me question the validity of table 1 as well. I hope the author can clarify this in the rebuttal.

One experiment that is missing and could have been interesting is to study the performance of the algorithm depending on the quality of the heuristic.

I suggest merging the related work section and section 3.3.

**Questions:**

-	Why doesn’t the algorithm need more data since you need to collect data with both policies?
-	Equation (7) and (8) introduce an off-policy aspect to the algorithm, is convergence still guaranteed?
-	Why use J_random in the formula to measure performance? Why not use a percentage of improvement over $J_{H-only}$?
-	What happened with H4-9-10-12 in figure 2?
-	On the same problem with different heuristic, how would the performance of HEPO look like depending on the asymptotic performance of $\pi_H$

**Limitations:**

The current algorithm seems limited to on policy methods. It also applies to problems where we can make a clear distinction between task reward and heuristic rewards.

---

> ### Author Rebuttal · Authors · 2024-08-07
>
> We sincerely appreciate the reviewer’s positive feedback on the simplicity of our method, the design and statistical rigor of our experiments, and the clarity of our writing. We have addressed the remaining questions below.
>
> > Why doesn’t the algorithm need more data since you need to collect data with both policies?
> >
>
> Each policy collects half the total number of trajectories for training. Suppose we aim to gather $B$ trajectories per training epoch to update both policies $\pi$ and $\pi_{\text{H}}$. HEPO collects $B/ 2$ trajectories using $\pi$ and $\pi_{\text{H}}$, respectively, and updates both policies with these $B$ trajectories.
>
> > Theoretical properties of Eq 7 & 8
> >
>
> HEPO's policy update objectives for $\pi$ and $\pi_H$ (Eq. 7 & 8) are derived from PPO's policy improvement objective. The key difference is that HEPO uses data collected by both $\pi^{i-1}$ and $\pi_H^{i-1}$ from the previous epoch $i-1$, while PPO uses data from only one policy $\pi^{i-1}$. More details are in Appendix A. This does not invalidate PPO's policy improvement objective because applying PPO's clipping trick to these equations (as done in our implementation) ensures that the updated policies remain close, preserving the validity of the PPO update. Consequently, HEPO's convergence can be analyzed by recent theoretical work (https://openreview.net/pdf?id=uznKlCpWjV) on PPO's convergence properties with neural network parameterization.
>
> > Limitation on convergence to optimal policy
> >
>
> We'd like to elaborate this limitation below: Solving HEPO’s constrained objective in Equation 3 of our manuscript,
>
> $$
> \max_{\pi} J(\pi) + H(\pi) ~ \text{subject to} ~ J(\pi) \geq J(\pi_{H}),
> $$
>
> doesn't guarantee that $\pi$ will converge to the optimal policy for $J$ since $\pi$ is maximizing the biased objective $J(\pi) + H(\pi)$.
>
> However, this issue can be resolved by reshaping the heuristic rewards $H$ using potential-based reward shaping (PBRS) for HEPO. PBRS ensures that $\text{arg}\max_\pi J(\pi) + H(\pi) = \text{arg}\max_\pi J(\pi)$, guaranteeing the modified HEPO’s optimization objective is unbiased.
>
> While training policies with PBRS often perform worse empirically than training with the heuristic objective $J(\pi) + H(\pi)$ (as shown in Section 4.1), our results in Figure 9 of the rebuttal PDF show that combining PBRS with HEPO yields a higher probability of improvement over the policy trained with the heuristic objective (though still worse than HEPO trained without PBRS). This indicates that combining PBRS and HEPO is promising for achieving both good empirical performance and theoretical guarantees.
>
> > Comparison with EIPO
> >
>
> Though EIPO and HEPO share a similar formulation, we didn’t include EIPO in the experiments because its purpose is not to enhance performance with heuristic rewards but to ensure that policies trained with heuristics do not degrade the performance of policies trained solely with task rewards.
>
> That being said, in Figure 8 of the rebuttal PDF, we compare EIPO and HEPO using the Isaac and Bi-Dex benchmarks (as done in the experiments in Section 4.1). The results indicate that HEPO outperforms EIPO in both IQM of return and probability of improvement. EIPO is implemented according to the original EIPO paper, collecting trajectories by alternating between two policies and training the reference policy solely with task rewards.
>
> > The current algorithm seems limited to on policy methods.
> >
> We show that HEPO is also effective at off-policy RL, too. Please see "Generality on SAC" in the general response.
>
> > Baselines in Table 1
> >
>
> We didn’t include HuRL and PBRS in Table 1 due to their poor performance in Figure 1. In Table 3 of our rebuttal PDF, we add HuRL and PBRS for comparison, showing that both perform poorly.
>
> > Quality of the heuristic/asymptotic performance v.s. HEPO performance
> >
>
> We believe that Figure 2 reveals the relationship between HEPO's performance and the quality of the heuristic reward. The policy trained with only heuristic rewards (H-only) represents both the asymptotic performance of $\pi_H$ in HEPO and the quality of the heuristic itself. We found a positive correlation (Pearson coefficient of 0.9) between the average performances of H-only and HEPO in Figure 2's results, suggesting that better heuristics lead to improved HEPO performance. Figure 7(c) in our rebuttal PDF provides more details.
>
> > Related work and section 3.3.
> >
>
> Thanks! We will merge them.
>
> > Why not use a percentage of improvement over $J_{H-only}$
> >
> The formula $(J - J_{\text{min}}) / (J_{\text{max}} - J_{\text{min}})$ is a standard method to normalize scores in deep RL [1]. Here, we define $J_{\text{random}} = J_{\text{min}}$ and $J_{\text{Honly}} = J_{\text{max}}$.
>
> **Why not percentage of improvement $J / J_{\text{max}}$?** It's because $J_{\text{max}}$ can be negative. For instance, if $J_{\text{max}} = -2$ and the performance of algorithms A and B are 1 and 10, the normalized scores are $1 / -2 = -0.5$ for A and $10 / -2 = -5$ for B, respectively. This incorrectly suggests algorithm A is better than B despite A's score (1) being worse than B's (10). Subtracting $J_{\text{min}}$ from both $J$ and $J_{\text{max}}$ ensures both the numerator and denominator are positive, addressing this issue.
>
> [1] Deep reinforcement learning at the edge of the statistical precipice, NeurIPS 2021
>
> > H4-9-10-12 in figure 2
> >
>
> As mentioned in Section 4.2 of our manuscript, the heuristic reward functions h12 and h9 had an incorrect sign for the distance component. This error caused the policy to reward moving away from the cabinet instead of toward it, making the learning task more difficult.
>
> **H4 & H10:** Both heuristic reward functions include multiple axis alignment rewards, like aligning the robot's end-effector with the target cabinet's forward axis. The poor performance likely stems from a designer misunderstanding the coordinate system, resulting in rewards that direct the robot away from the cabinet.

---

> > ### Comment · Reviewer_P4zd · 2024-08-08
> > **Thank you for the detailed response and the additional experiments.**
> >
> > I think you answered all of my questions quite clearly.
> >
> > I would advise to modify the table 1 from the paper with the full results and add the plots from the rebuttal to the appendix as they are quite useful in understanding the performance of the method. I will update my score.

---

### Official Review · Reviewer_15cn · 2024-07-09

**Soundness:** 3
**Presentation:** 3
**Contribution:** 3
**Rating:** 7
**Confidence:** 3

**Summary:**

This paper introduces a constrained optimization approach for reinforcement learning, ensuring that the learned policy outperforms or matches the performance of policies trained solely with heuristic rewards. In simulations, the proposed HEPO method consistently delivers superior task performance across various tasks, demonstrating improved returns in robotic locomotion, helicopter, and manipulation tasks.

**Strengths:**

This paper is straightforward and easy to follow, with a clearly stated motivation. It introduces an innovative constrained optimization approach that effectively leverages heuristic signals to enhance task performance in reinforcement learning. The effectiveness of the HEPO method is demonstrated through its consistent superiority across diverse tasks such as robotic locomotion, helicopter control, and manipulation, even with limited training data. Additionally, the proposed method simplifies the reward design process by dynamically balancing heuristic and task rewards without the need for manual tuning, showing significant improvements over traditional heuristic-only and mixed reward approaches.

**Weaknesses:**

The paper would benefit from a more thorough theoretical analysis. Including a theoretical guarantee on the convergence rate, an examination of how the selection of heuristics impacts the HEPO method, and an analysis of the effect of the alpha parameter would enhance the robustness and depth of the study.

**Questions:**

Could you please explain how the heuristic is obtained in the implementation? Additionally, it would be helpful to see the computational cost of solving the constrained optimization problem. Is it possible to include a comparison of your method with reward shaping and policy cloning methods? For equation 5, when rearranging equation 4, could you clarify where the term J(pi_H) goes?

**Limitations:**

While the HEPO method shows empirical success, it lacks a thorough theoretical analysis, including a formal guarantee on the convergence rate and the effects of heuristic selection and the alpha parameter on performance. Additionally, the computational cost of solving the constrained optimization problem is not addressed, which could impact the feasibility of the approach in large-scale or real-time applications. Furthermore, the study does not compare HEPO with other reinforcement learning techniques such as reward shaping and policy cloning, which would provide a more comprehensive evaluation of its effectiveness.

---

> ### Author Rebuttal · Authors · 2024-08-07
>
> We're thrilled that the reviewer love our paper and appreciate their compliments on the clarity of our presentation, the novelty of our proposed method, and the thoroughness of our experimental evaluation. The following are our responses to the remaining comments:
>
> > theoretical guarantee on the convergence rate
> >
>
> HEPO can be considered a constrained RL problem, where the constraint is that the policy’s expected return must improve over the reference policy (i.e., $J(\pi) \geq J(\pi_{\text{ref}})$). Therefore, existing theoretical analyses of convergence rates (https://arxiv.org/abs/2407.10775) for constrained RL with expected value as constraints can be applied to HEPO.
>
> > examination of how the selection of heuristics impacts the HEPO method
> >
>
> At the core of HEPO is training policies by RL with heuristic rewards, which the impact of heuristic selection has been analyzed theoretically in previous work (https://arxiv.org/abs/2210.09579). Empirically, Figure 2 in our manuscript compares HEPO and PPO using different heuristic rewards for the same task reward. The results show that HEPO is less sensitive to heuristic reward selection and outperforms PPO on average. Also, Figure 7(c) in the rebuttal PDF shows that HEPO’s performance scales with the quality of the heuristic (i.e., the performance of training with heuristic rewards only).
>
> > an analysis of the effect of the alpha parameter
> >
>
> In Appendix B.2.2, we analyzed the effect of the alpha parameter and showed that HEPO is not sensitive to its learning rate.
>
> > Could you please explain how the heuristic is obtained in the implementation?
> >
>
> As outlined at the beginning of Section 4 in our manuscript, our heuristics are based on the source code from Isaac Gym [1] and Bidexterous Manipulation [2]. Further details are available in their respective papers and repositories. In Section 4.2, we use heuristic reward functions designed by human subjects to evaluate the performance of each method on reward functions of varying quality. More information can be found in Section 4.2.
>
> [1] Makoviychuk, Viktor, et al. "Isaac gym: High performance gpu-based physics simulation for robot learning." *arXiv preprint arXiv:2108.10470* (2021).
>
> [2] Chen, Yuanpei, et al. "Towards human-level bimanual dexterous manipulation with reinforcement learning." *Advances in Neural Information Processing Systems* 35 (2022): 5150-5163.
>
> > Additionally, it would be helpful to see the computational cost of solving the constrained optimization problem.
> >
>
> Updating the constrained problem in HEPO requires only an additional memory allocation to store the Lagrangian multiplier $\alpha$. After finishing one epoch of policy updates, we perform a single gradient step to update $\alpha$, resulting in only a minor computational overhead.
>
> > Is it possible to include a comparison of your method with reward shaping and policy cloning methods?
> >
>
> Please note that Section 4.1 of our manuscript provided a comparison of well-known reward shaping methods (PBRS [1] and HuRL [2]) and showed that HEPO outperforms both methods.
>
> Regarding policy cloning, we kindly ask the reviewers to clarify the definition of policy cloning. If it refers to behavior cloning that is trained to imitate expert demonstration, this might be outside the scope of our work, as we do not assume a dataset of expert demonstrations. Instead, our focus is on improving RL algorithms' performance when trained with heuristic reward functions.
>
> [1] Ng, Andrew Y., Daishi Harada, and Stuart Russell. "Policy invariance under reward transformations: Theory and application to reward shaping." *Icml*. Vol. 99. 1999.
>
> [2] Cheng, Ching-An, Andrey Kolobov, and Adith Swaminathan. "Heuristic-guided reinforcement learning." *Advances in Neural Information Processing Systems* 34 (2021): 13550-13563.
>
> > For equation 5, when rearranging equation 4, could you clarify where the term J(pi_H) goes?
> >
>
> $J(\pi_H)$ can be considered a constant for the optimization in Equation 5 since it doesn’t influence the gradient of policy $\pi$. Therefore, it can be omitted in Equation 5.

---

> > ### Comment · Reviewer_15cn · 2024-08-08
> > **Thank you for the timely response**
> >
> > Your answers addressed my questions. I have no further comments.

---

### Official Review · Reviewer_inih · 2024-07-12

**Soundness:** 3
**Presentation:** 3
**Contribution:** 2
**Rating:** 6
**Confidence:** 3

**Summary:**

The paper presents a novel approach to improve reinforcement learning (RL) by incorporating heuristic signals. The authors propose a constrained optimization method that uses heuristic policies as references to ensure that the learned policies outperform heuristic policies on the exact task objective. The method, named Heuristic-Enhanced Policy Optimization (HEPO), was tested on various robotic tasks, demonstrating consistent performance improvements over standard heuristic methods.

**Strengths:**

- The paper introduces a simple method to improve reinforcement learning (RL) algorithms using heuristic rewards, by shifting focus from ensuring optimal policy invariance to simply outperforming heuristic policies.
- The method is validated on various environments, showing consistent performance improvements.
- The paper is clearly written.

**Weaknesses:**

- The claim that the contribution is an add-on to existing deep RL algorithms is not fully substantiated, as the authors only evaluate HEPO using Proximal Policy Optimization (PPO). Demonstrating HEPO's performance with other deep RL algorithms, such as Soft Actor-Critic (SAC), would help validate this claim.
- The method is very similar to Extrinsic-Intrinsic Policy Optimization (EIPO), with the main difference being the reference policy used. The authors should provide more details on this difference, including the number of additional hyperparameters introduced in EIPO and a comparison of its complexity in terms of time with HEPO.
- The benchmarks used in this paper differ from those in previous works like EIPO [6] and HuRL [7]. The authors should address how HEPO would perform on those benchmarks to ensure a fair comparison.
- The paper does not include challenging tasks such as Montezuma's Revenge.
- It is unclear whether the reward functions and heuristics used in the experiments are designed by the authors or sourced from ISAAC and BI-DEX benchmarks.

**Questions:**

- Please check weaknesses.
- How does HEPO perform with other deep RL algorithms, such as Soft Actor-Critic (SAC)?
- What specific heuristics are used in the progressing tasks, and how are the rewards sparse/delayed?
- In figure3: Is the T-only ablation equivalent to the EIPO algorithm or a modified version of HEPO?
- Line 129 is missing a reference to the application of PPO in robotics with heuristic rewards.

**Limitations:**

The authors address the limitations and potential negative impacts in their paper.

---

> ### Author Rebuttal · Authors · 2024-08-07
>
> We sincerely thank the reviewer for appreciating our idea of shifting the focus of optimal policy invariance, as well as our extensive evaluation and clear writing. We address the remaining comments below:
>
> > Q1: Demonstrating HEPO's performance with other deep RL algorithms, such as Soft Actor-Critic (SAC), would help validate this claim.
> >
>
> We integrated HEPO into HuRL's SAC codebase and found that HEPO outperforms SAC and matches HuRL's performance on the most challenging task (see Figure 7(b) in the rebuttal PDF). HuRL used optimized hyperparameters for this task, as reported in its paper, while HEPO used the same hyperparameters as in Section 4.1 of our manuscript. Other tasks in HuRL's paper require pre-trained value functions as heuristic reward functions, but the checkpoints were unavailable online.
>
> > Q2: The method is very similar to Extrinsic-Intrinsic Policy Optimization (EIPO), with the main difference being the reference policy used. The authors should provide more details on this difference, including the number of additional hyperparameters introduced in EIPO and a comparison of its complexity in terms of time with HEPO.
> >
>
> We acknowledged the similarities between EIPO and HEPO and discussed the differences in Section 3.3 of our manuscript. The difference in the choice of reference policy doesn’t add more hyperparameters or time complexity.
>
> **Hyperparameters:** Both HEPO and EIPO share the same number of  hyperparameters: 1. Initial value of the Lagrangian multiplier and 2. Learning rate of the Lagrangian multiplier.
>
> **Time complexity:** It’s the same for both methods:
>
> - **Trajectory rollout:** Both rollout $B$ trajectories (number of parallel environments), involving the same number of forward passes on the policy networks. Thus, both have the same time complexity in this stage.
> - **Policy update:** Both train two policies on the collected trajectories at each iteration, so HEPO does not introduce additional complexity during training.
>
> > Q3: The benchmarks used in this paper differ from those in previous works like EIPO [6] and HuRL [7]. The authors should address how HEPO would perform on those benchmarks to ensure a fair comparison. The paper does not include challenging tasks such as Montezuma's Revenge.
> >
>
> HEPO outperformed both HuRL and EIPO on the most challenging tasks reported in their papers. For HuRL, see Q1. For EIPO, we tested both methods on Montezuma's Revenge using RND exploration bonuses (also used in EIPO's paper), which are crucial for good performance on this task. Figure 7(a) in the rebuttal PDF shows that HEPO significantly outperforms EIPO, with no overlap in their confidence intervals at the end of training. Additionally, HEPO achieves the same performance as training PPO with RND bonuses at convergence (2 billion frames), as reported in RND's paper, while using only 20% of the training data, demonstrating significantly improved sample efficiency.
>
> > Q4: It is unclear whether the reward functions and heuristics used in the experiments are designed by the authors or sourced from ISAAC and BI-DEX benchmarks.
> >
>
> As described at the beginning of Section 4 in our manuscript, our heuristic reward functions are all based on the source code from Isaac Gym [1] and Bidexterous Manipulation [2]. More details can be found in their respective papers and repositories.
>
> [1] Makoviychuk, Viktor, et al. "Isaac gym: High performance gpu-based physics simulation for robot learning." arXiv preprint arXiv:2108.10470 (2021).
>
> [2] Chen, Yuanpei, et al. "Towards human-level bimanual dexterous manipulation with reinforcement learning." Advances in Neural Information Processing Systems 35 (2022): 5150-5163.
>
>
> > Q5: What specific heuristics are used in the progressing tasks, and how are the rewards sparse/delayed?
> >
>
> In progressing tasks, large rewards can be sparse or delayed because making progress often requires actions that don't immediately advance the robot. For example, in Humanoid tasks, the robot must balance itself to move quickly and get large rewards. However, achieving balance requires a sequence of actions, which delays the large reward associated with fast movement.
>
> > Q6: In figure3: Is the T-only ablation equivalent to the EIPO algorithm or a modified version of HEPO?
> >
>
> "HEPO ($\pi_{\text{ref}}$ = T-only)" is an ablated version of HEPO whose reference policy is trained solely with task rewards. This differs from EIPO only in the trajectory collection method, as shown in Figure 3(b). To clarify, "T-only" should be changed to "J-only," indicating the policy is trained only with task rewards. We will correct this typo in the next manuscript.
>
> > Q7: Line 129 is missing a reference to the application of PPO in robotics with heuristic rewards.
> >
>
> Thanks for pointing this out. We plan to add the following references on robotics [1, 2].
>
> [1] Chen, Tao, et al. "Visual dexterity: In-hand reorientation of novel and complex object shapes." *Science Robotics* 8.84 (2023)
>
> [2] Lee, Joonho, et al. "Learning quadrupedal locomotion over challenging terrain." *Science robotics* 5.47 (2020)

---

> > ### Comment · Reviewer_inih · 2024-08-11
> >
> > Thank you for your responses. I have updated my rating.

---

### Author Rebuttal · Authors · 2024-08-07

We're delighted that the reviewers find our manuscript easy to follow. We sincerely thank them for appreciating the simplicity of our method and the breadth of our experiments. Here, we'd like to summarize the common questions and the new experiments:

### Additional Results

- **Generality on SAC (`#inih`):** We integrated HEPO into HuRL's SAC codebase. Despite HuRL using tuned hyperparameters as reported in its paper [1], HEPO outperformed SAC and matched HuRL on the most challenging task (Figure 7(b) in the rebuttal PDF) using the same hyperparameters from Section 4.1 of our manuscript, showing the generality of HEPO on different RL algorithms.
- **Better than EIPO on the most challenging task (`#inih`):** Comparing HEPO and EIPO on the most challenging task, Montezuma's Revenge, reported in EIPO's paper [2] using RND exploration bonuses [3] (as used in the EIPO paper), Figure 7(a) in the rebuttal PDF shows that HEPO performs better than EIPO. Also, HEPO matched PPO trained with RND bonuses at convergence (2 billion frames) reported in [3] using only 20% of the training data, demonstrating drastically improved sample efficiency.
- **Additional Baselines (`#P4zd`):**
    - **Outperforming EIPO in IsaacGym + Bi-Dex (Section 4.1):** Figure 8 in the rebuttal PDF shows that HEPO outperforms EIPO in both IQM of return and probability of improvement using Isaac [4] and Bi-Dex [5] benchmarks. EIPO was implemented per the original paper [2], alternating between two policies and training the reference policy solely with task rewards.
    - **Outperforming baselines in the experiments of human-designed heuristics in the wild (Section 4.2):** Table 3 in our rebuttal PDF shows HuRL and PBRS perform poorly with human-designed reward functions from Table 1 in real-world scenarios.
- **Synergistic with potential-based reward shaping (PBRS) [6] (`#P4zd`):** By reshaping heuristic rewards $H$ using PBRS for HEPO, we ensure $\text{arg}\max_\pi J(\pi) + H(\pi) = \text{arg}\max_\pi J(\pi)$, guaranteeing the modified HEPO’s optimization objective is unbiased. While training with PBRS often performs worse than the policy trained with $J(\pi) + H(\pi)$ (see Section 4.1), our results in Figure 9 of the rebuttal PDF show that combining HEPO with PBRS improves the probability of improvement over nine IsaacGym tasks. This suggests that combining PBRS and HEPO is promising for both empirical performance and theoretical guarantees in optimizing an unbiased objective that yields the same optimal policy for $J$ asymptotically.

### Clarification

- **Source of heuristic reward functions (`#inih/#15cn`):** Our heuristic reward functions are based on the source code of Isaac Gym [4] and Bidexterous Manipulation [5], as mentioned at the beginning of Section 4 in our manuscript. For more information, please refer to their respective papers and repositories. In the experiments on reward functions designed in the wild (Section 4.2), humans were recruited to write the heuristic reward functions. See Section 4.2 for details.
- **EIPO and HEPO have the same memory/time complexity (`#inih/#P4zd/#15cn`):**
    - **Hyperparameters/Memory:** Both EIPO and HEPO share the same number of hyperparameters: 1. Initial value of the Lagrangian multiplier, and 2. Learning rate of the Lagrangian multiplier. Thus, HEPO does not require additional memory or hyperparameters compared to EIPO.
    - **Time Complexity:**
        - **Trajectory Rollout:** Both rollout $B$ trajectories (number of parallel environments), thus involving the same number of forward passes on the policy networks and resulting in the same time complexity at this stage.
        - **Policy Update:** Both train two policies on the collected trajectories at each iteration, so HEPO does not add complexity during training.

[1] Cheng, Ching-An, Andrey Kolobov, and Adith Swaminathan. "Heuristic-guided reinforcement learning." Advances in Neural Information Processing Systems 34 (2021): 13550-13563.

[2] Chen, Eric, et al. "Redeeming intrinsic rewards via constrained optimization." Advances in Neural Information Processing Systems 35 (2022): 4996-5008.

[3] Burda, Yuri, et al. "Exploration by random network distillation." arXiv preprint arXiv:1810.12894 (2018).

[4] Makoviychuk, Viktor, et al. "Isaac gym: High performance gpu-based physics simulation for robot learning." arXiv preprint arXiv:2108.10470 (2021).

[5] Chen, Yuanpei, et al. "Towards human-level bimanual dexterous manipulation with reinforcement learning." Advances in Neural Information Processing Systems 35 (2022): 5150-5163.

[6] Ng, Andrew Y., Daishi Harada, and Stuart Russell. "Policy invariance under reward transformations: Theory and application to reward shaping." *Icml*. Vol. 99. 1999.

---

> ### Author Response · Authors · 2024-08-07
> **update the figure**
>
> We missed one curve in the rebuttal PDF and have updated Figure 7(a) in the following link: https://imgur.com/jzxWoq6. Note that we didn't train EIPO and HEPO for 2 billion frames since it will take around two weeks to finish.

---

### Decision · Program_Chairs · 2024-09-25

**Decision:**

Accept (poster)

**Comment:**

The paper proposes a method, HEPO, for ensuring that heuristics help RL by making RL optimize a constraint objective, where the constraint is given by the value function of a policy trained on heuristic rewards. HEPO is closely related to an existing approach, EIPO, but provides valuable improvements over it in implementation simplicity and reduced number of hyperparameters. The reviewers found the paper easy to read and suggested a number of improvements to the experiments, such as using SAC, that the authors have successfully implemented during the rebuttal stage. The reviewers and the meta-reviewers trust that the authors will include them into the manuscript.

After these new results are incorporated into the paper, the work's only notable remaining weakness (which is addressed in the rebuttals only partly) is the scarcity of its theoretical analysis. However, the metareviewer believes that the paper's algorithmic contributions are sufficiently valuable on their own and fully recommends this paper for acceptance.